# Identification of a potent V3 glycan site broadly neutralizing antibody targeting an N332$_{gp120}$ glycan-independent epitope

Broadly neutralizing antibodies (bNAbs) against HIV-1 can suppress viremia in vivo and inform vaccine development. Here we characterized 007, a V3 glycan site bNAb exhibiting high levels of antiviral activity against multiclade pseudovirus panels. 007 targets an N332$_{gp120}$ glycan-independent V3 epitope, a site of the HIV-1 envelope protein (Env) vulnerability to which only weakly neutralizing antibodies had previously been identified. Functional analyses demonstrated distinct binding and neutralization profiles compared to classical V3 glycan site bNAbs. A 007 Fab-Env cryogenic electron microscopy structure revealed contacts with the V3 $^{324}$GD/NIR$^{327}$ motif and interactions with N156$_{gp120}$ and N301$_{gp120}$ glycans. In contrast to classical V3 bNAbs, 007 binding to Env does not depend on the N332$_{gp120}$ glycan, rendering it resistant to common escape mutations. Structures of 007 IgG-Env trimer complexes showed two Env trimers crosslinked by three bivalent IgGs. Bivalent 007 IgG was more potent than monovalent 007 IgG heterodimer, suggesting a role for avidity in potent neutralization. Finally, in HIV-1$_{ADA}$-infected humanized mice, 007 caused transient decline of viremia and overcame classical V3 escape mutations, highlighting 007's potential for HIV-1 prevention, therapy, functional cure and vaccine design.

Broadly neutralizing antibodies (bNAbs) targeting the HIV-1 envelope protein (Env) inform vaccine design, hold potential for therapy and prevention and advance efforts toward achieving a functional cure[1]. However, clinical trials have underscored the stringent requirements for enhanced antiviral activity that will be critical to counteract virus Env diversity and emergence of escape. Combined application of bNAbs with complementary neutralization coverage offers an opportunity to overcome these challenges[2]. Thus, the discovery of new bNAbs demonstrating distinct binding modes, neutralizing profiles and viral escape pathways remains essential to facilitate successful clinical application of bNAbs.

On the HIV-1 Env trimer, bNAbs recognize highly conserved epitopes essential for viral entry[3–8]. One such epitope is a V3 glycan site located at the base of the V3 loop. This epitope includes the N332$_{gp120}$ glycan, the $^{324}$GD/NIR$^{327}$ motif and N-linked glycans in the vicinity (N133$_{gp120}$, N137$_{gp120}$, N156$_{gp120}$ and N301$_{gp120}$)[9]. However, some V3 glycan site bNAbs are also known to exhibit promiscuity in their glycan recognition and/

or accommodation, allowing tolerance to shifts in N-glycan composition and configuration. For example, although bNAbs such as 10-1074, BG18, PGT124 and DH270 are highly dependent on the presence of the N332$_{gp120}$ glycan, others like PGT121, PGT128 and PGT130 can compensate for the loss of the N332$_{gp120}$ glycan by targeting alternative glycans within the high-mannose patch[9–12]. Recently, a neutralizing antibody (nAb), EPTC112, that lacks contacts with the N332$_{gp120}$ glycan and instead targets a previously undescribed N332$_{gp120}$ glycan-independent V3 epitope extending to glycans of the V1 loop was reported[13]. However, EPTC112 displayed low levels of breadth (23%, 142 virus strains) and potency (GeoMean IC$_{50}$ against all strains = 2.6 µg ml$^{-1}$), limiting its applicability for prevention and immunotherapy.

Anti-HIV-1 bNAbs neutralize the virus through multiple mechanisms, including blocking of receptor binding, hindering membrane fusion and accelerating decay of Env trimers[3]. As with antibodies to other antigens[14], immunoglobulin G (IgG) bivalency could enhance the breadth and potency of HIV-1 bNAbs by permitting the binding of

✉e-mail: florian.klein@uk-koeln.de

adjacent Envs on the surface of the virus (inter-spike crosslinking) or simultaneously engaging with multiple protomers of the same Env (intra-spike crosslinking)[15,16]. However, HIV-1 bNAb IgGs are usually not known to utilize bivalent binding, a likely consequence of the relatively few Envs coating the surface of the virus and the positioning of conserved bNAb epitopes preventing inter- and intra-Env crosslinking, respectively[15]. Thus, the discovery of naturally occurring bNAbs that utilize avidity presents an opportunity for optimizing prevention and immunotherapy, as well as for informing vaccine design.

Here, we report on the identification and detailed characterization of the anti-HIV-1 bNAb 007 targeting an $N332_{gp120}$ glycan-independent V3 epitope of the Env trimer. Cryogenic electron microscopy (cryo-EM) analyses revealed a distinct binding mode compared to canonical V3 glycan site bNAbs that depends on the $N301_{gp120}$ and $N156_{gp120}$ glycans. Bivalent 007 IgG was more potent than its monovalent forms, and a structure of 007 IgG in complex with SOSIP trimers revealed a stable dimer of Env trimers linked by IgGs. Moreover, 007 displayed high levels of antiviral activity and exhibited a different neutralization profile compared to V3 reference bNAbs against extended multiclade panels. In HIV-1$_{ADA}$-infected humanized mice, 007 achieved transient decline of viremia and overcame V3 escape mutations. In conclusion, our findings suggest the $N332_{gp120}$-independent V3 epitope as a viable target for vaccine design and expand the armamentarium of bNAbs available for clinical use.

## Results

### Identification of bNAb 007 with a distinct binding and neutralizing profile

Donor EN01 was identified as the top elite neutralizer from Mbeya region in southern Tanzania which was part of a multinational cohort of 2,354 people living with HIV (PLWH)[17]. Purified serum IgG from donor EN01 displayed the highest levels of breadth (100%) and potency (mean neutralization of 99% at 300 μg ml$^{-1}$) against the 12-strain global HIV-1 pseudovirus panel[17,18] (Fig. 1a and Supplementary Table 1). Neutralization fingerprint analyses to determine the potential epitope specificity of the serum IgG response were inconclusive due to ambiguous delineation scores[17,19] (Supplementary Table 1). To identify nAbs mediating the broad and potent serum response, we isolated single HIV-1 Env-reactive B cells using a GFP-labeled BG505$_{SOSIP.664}$ bait protein[20]. HIV-1 Env-reactive B cells were isolated at a frequency of 0.46%, and a total of 189 heavy chain and 100 light chain sequences were amplified using optimized PCR primers and protocols[21] (Extended Data Fig. 1a). Heavy chain sequence analyses revealed a high degree of clonality among the isolated BCR sequences, identifying 23 B cell clones with two or more members. Compared to a human reference memory IgG repertoire dataset[22], amplified BG505$_{SOSIP.664}$-reactive V$_H$ sequences of donor EN01 displayed higher levels of somatic mutation (median V$_H$ gene germline identity of 85.3% versus 95.3% on nucleotide level), an enrichment of V$_H$ gene segments 1-69 and 4-4, and comparable CDRH3 lengths (Extended Data Fig. 1a, b).

A total of 48 representative mAbs were produced and evaluated for neutralizing activity at a concentration of 2 μg ml$^{-1}$ against a panel of six HIV-1 pseudoviruses representing different clades (Extended Data Fig. 1c). Among the screened mAbs, 007 identified from B cell clone 17 neutralized all strains of the screening panel with a mean neutralization of 90% at 2 μg ml$^{-1}$. Members of this clone derive from V$_H$4-34*01/02 and V$_κ$1-12*01 gene segments with CDRH3 and CDRL3 lengths of 22 and 9 amino acids (Supplementary Table 2) and belong to the IgG3 subclass. Unless otherwise specified, mAbs were expressed and analyzed as IgG1. V$_H$ gene germline identities from this clonal family ranged from 75.0% to 80.5%, whereas V$_κ$ gene germline identities ranged from 76.2% to 85.0% at the nucleotide level (Supplementary Table 2). All members of B cell clone 17 exhibited high antiviral activity against the 12-strain global HIV-1 pseudovirus panel[18], with 007 being the most potent antibody of the clone (breadth = 100%; GeoMean IC$_{50}$/

IC$_{80}$ = 0.008/0.069 μg ml$^{-1}$) (Fig. 1b and Supplementary Table 3). Notably, the neutralizing activity of 007 and serum IgG from donor EN01 showed only a moderate correlation across a panel of 40 pseudoviruses (Pearson $r$ = 0.54), indicating that members of the 007 clonal lineage contribute to, but do not fully account for, the donor's serum neutralizing activity (Extended Data Fig. 1d, e). To map the epitope of 007, we performed competition ELISAs using the BG505$_{SOSIP.664}$ Env trimer containing the T332N amino acid substitution. We detected interference with the CD4bs bNAbs 1-18 and 3BNC117 as well as with the V3 glycan site bNAb 10-1074 (ref. 10) (Fig. 1c). However, although 007 competed with 3BNC117 and 1-18 in Env trimer binding, no reduction in neutralizing activity was detected when these antibodies were combined in traditional neutralization assays indicating functional compatibility (Supplementary Table 4). In addition, we determined the neutralizing activity of 007 against BG505$_{T332N}$ pseudovirus mutants that abrogate the activity of V3 glycan site, CD4bs, V2 apex, MPER, and silent face reference bNAbs. None of the tested signature escape mutations reduced the neutralizing activity of 007 (Supplementary Table 5). To delineate the neutralization profile of 007, we evaluated the antiviral activity against the f61 fingerprint[19] and 119 multiclade pseudovirus panel[23] (Supplementary Tables 6 and 7). Against these panels, the neutralizing activity of 007 did not show a consistent correlation or alignment with reference bNAbs from known epitope classes (Fig. 1d, e). Next, we evaluated the potential reactivity of 007 to self-antigens using a HEp-2 staining assay. Unlike the anti-HIV-1 bNAb 2F5, which exhibits known autoreactivity[24], 007 showed no detectable autoreactivity against HEp-2 cells (Extended Data Fig. 2a). However, 007 demonstrated a pharmacokinetic profile less favorable to that of bNAbs 10-1074 and 3BNC117 in human FcRn transgenic mice (B6.Cg-Fcgrt$^{tm1Dcr}$ Prkdc$^{scid}$Tg(FCGRT)32Dcr/DcrJ, $n$ = 4, all female), which serve as reference for IgG1 half-lives in humans (Extended Data Fig. 2b). These findings indicate that 007 achieves high antiviral activity without evidence of autoreactivity and targets an epitope distinct from previously characterized bNAbs.

### 007 targets an $N332_{gp120}$ glycan-independent V3 epitope on Env

To elucidate the binding mode of 007, we performed single-particle cryo-EM analysis of the antibody Fab in complex with a soluble BG505 Env trimer, which included SOSIP stabilizing mutations[20] and an engineered disulfide (I201C$_{gp120}$-A433C$_{gp120}$) designed to prevent trimer opening (BG505-DS). Fab was added in a greater than threefold molar excess to SOSIP trimer, and Fab-Env complexes were then isolated by size exclusion chromatography (SEC). Structural analysis revealed four trimer classes with 0, 1, 2 or 3 bound Fabs, with the most populated class being a 3.0 Å single Fab-bound trimer (Fig. 2a, Extended Data Fig. 3a and Supplementary Table 8). Comparison of the 007 binding pose to poses of V3 glycan site bNAbs showed a distinct mode of binding, most closely resembling the V3 glycan site bNAb PGT128 and EPTC112 (Fig. 2b and Extended Data Fig. 3b).

The structure revealed 007 is framed by N-linked glycans attached to N156$_{gp120}$ and N301$_{gp120}$ but does not contact the glycan at N332$_{gp120}$, which is important for binding of many V3 glycan bNAbs[12,25,26] (Fig. 2c, d). Antibody contacts with protein portions of Env are mediated exclusively by a 22-residue CDRH3 which extends outward to contact the conserved $^{324}$GD/NIR$^{327}$ motif on Env (Fig. 2c, e). Within this region, F100E$_{HC}$ contacts G324$_{gp120}$, and E100D$_{HC}$ at the tip of the CDRH3 is in close proximity to R327$_{gp120}$ (Fig. 2e). Other V3 bNAbs possess negatively charged glutamate residues within their CDRH3s that may form electrostatic contacts with the positively-charged R327$_{gp120}$ (E100I$_{HC}$ in 10-1074 (ref. 25), PGT122 (ref. 26) and PGT124 (ref. 27 and E100G$_{HC}$ in BG18 (ref. 12)). In addition, K99$_{HC}$ of 007 forms an electrostatic interaction with D322$_{gp120}$ (D321(1)$_{gp120}$ in some numbering nomenclatures) within V3 (Fig. 2e and Extended Data Fig. 4a), a residue that is a negatively charged Asp or Glu in >70% of sequences (www.hiv.lanl.gov). The CDRH3 is

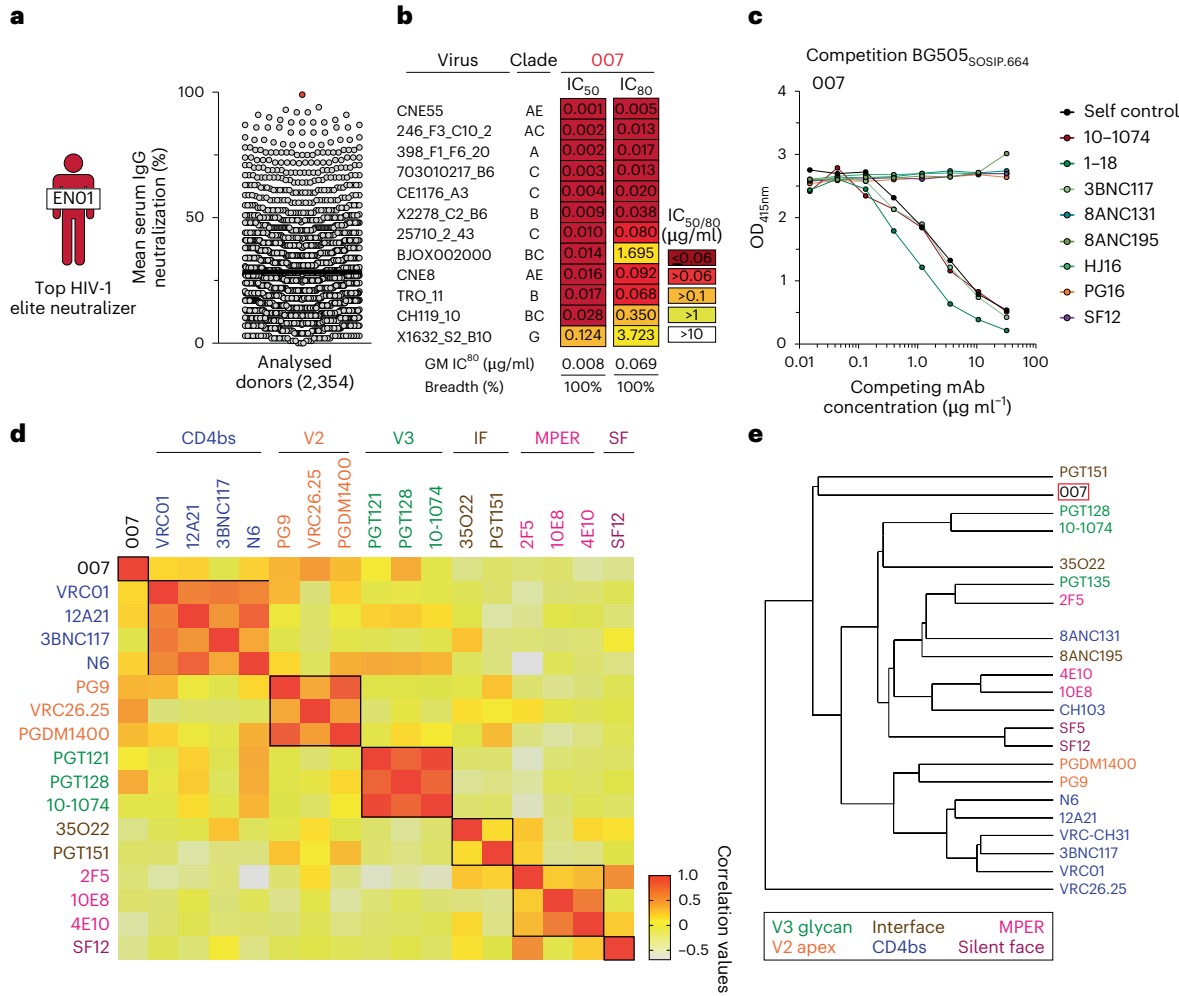

**Fig. 1 | Identification of bNAb 007 with distinct binding and neutralizing activity. a**, HIV-1 neutralizing serum activity of HIV-1 elite neutralizer EN01 against the global panel. Serum IgG samples were tested in duplicates[17,49]. **b**, Neutralization activity of bNAb 007 against the HIV-1 global pseudovirus panel. Samples were tested in duplicates. **c**, Interference of 007 with selected reference bNAbs targeting known epitopes on the HIV-1 Env trimer, as determined by competition ELISAs. **d**, Comparison of the neutralization profile of 007 with reference bNAbs targeting known epitopes against the f61 fingerprinting panel and **e**, 119 multiclade pseudovirus panel[23]. Antibody 007 was tested in duplicates. Neutralization data of reference bNAbs (**d** and **e**) were retrieved from CATNAP database.

also positioned near the gp120 V1 loop, which is located adjacent to V3, thereby allowing 007 $L100A_{HC}$ and $L100F_{HC}$ to contact $R151_{gp120}$ within V1 (Extended Data Fig. 4b). Interestingly, in gp120 protomers bound by 007, but not in unbound gp120s, a portion of the V1 loop is disordered (Extended Data Fig. 4c), suggesting that 007 binding destabilizes the V1 loop.

007 targets the V1/V3 epitope, closely resembling that of the recently described bNAb EPTC112 (ref. 13) but engages it with a distinct, rotated binding orientation (Extended Data Fig. 3b). Both antibodies depend on the $N156_{gp120}$ and $N301_{gp120}$ glycans, but not the $N332_{gp120}$ glycan, and both contact the $^{324}GD/NIR^{327}$ motif on Env. However, 007 exhibits greater breadth and is more potent (GeoMean $IC_{50}/IC_{80}$ = 0.01/0.03 versus 0.32/0.29 μg ml⁻¹, breadth = 66/54% versus 28/14%, 110 or 105 virus strains) than EPTC112 (Supplementary Table 7). Structurally, EPTC112 contacts extend only to $D325_{gp120}$ of the $^{324}GD/NIR^{327}$ motif, whereas 007 interacts with the entirety of the motif (Fig. 2e, f). More extensive contact with the $^{324}GD/NIR^{327}$ motif and better accommodation of the $N156_{gp120}$ and $N301_{gp120}$ glycans could contribute to the enhanced neutralization properties of 007 compared with EPTC112.

N-glycans comprise much of the 007 epitope (Fig. 2c, d). The FWRH1/CDRH1 of 007 contains a glycine-rich stretch of amino acids

(GVHGVGLGGSGWG; $G23_{HC}$ – $G35_{HC}$), which wrap around the core pentasaccharide of the $N156_{gp120}$ glycan (Extended Data Fig. 4d). Five of these glycine residues arose from somatic hypermutation (Fig. 2c), but only one is an improbable mutation, as defined by the ARMADiLLO web server[28]. A similar mechanism of glycan accommodation is utilized by the VRC01-class CD4-binding site bNAbs, which either acquire deletions or glycine substitutions in CDRL1 to accommodate the $N276_{gp120}$ glycan[29]. In addition, the 007 CDRH2 packs against the $N301_{gp120}$ glycan (Extended Data Fig. 4d). Although extensive EM density was observed for both glycans, density corresponding to a core fucose, a component of complex-type glycans, was not observed, and more density was seen for the $N156_{gp120}$ glycan (eight monosaccharide subunits modeled) than the $N301_{gp120}$ glycan (six monosaccharide subunits modeled) (Fig. 2g). In contrast, for unbound protomers, relatively little EM density was observed at these positions (zero to two monosaccharide subunits modeled), as is typically the case for glycans not stabilized by interactions with an antibody. A caveat of these observations is that single-particle cryo-EM analysis of glycans is limited due to their compositional and conformational heterogeneity.

To evaluate the role of N-glycan processing in 007 neutralization, we expanded our neutralization analysis to include pseudoviruses produced in the presence of kifunensine, an inhibitor that prevents

processing of high-mannose N-glycans to complex-type carbohydrates. In contrast to other V3-targeting bNAbs, but in common with EPTC112[13], 007 did not neutralize kifunensine-treated viruses (Fig. 2h, Supplementary Table 9), implying that the Man-9 (that is, Man9(GlcNAc)2) high-mannose glycan trimming by mannosidase I is required for potent neutralization. We also performed site-specific N-glycan analysis using quantitative mass spectrometry profiling to determine the specific glycoforms enriched in 007-bound BG505 SOSIPs compared to total BG505 SOSIPs. The potential N-linked glycosylation site (PNGS) at $N301_{gp120}$ in 007-bound BG505 showed an increase in Man-5 high-mannose glycans compared to the same site in the total BG505 sample (Fig. 2i), consistent with EM density and neutralization data (Fig. 2g, h, Supplementary Table 9). Unambiguous glycoform identification at position $N156_{gp120}$ was not possible, however, as both $N156_{gp120}$ and $N160_{gp120}$ PNGSs resided on the same glycopeptide (Extended Data Fig. 5).

## 007 bivalency is required for potent neutralization

The observation of sub-stoichiometric trimer binding by a bNAb Fab could result from a weak monovalent binding interaction, which is unexpected for a potent bNAb such as 007. Using a surface plasmon resonance (SPR)-based assay, we confirmed that the binding affinities of 007 Fab were weak for the BG505 (7.5 μM) and BG505-DS (5.7 μM) SOSIPs, in contrast to the V3 bNAb Fabs 10-1074 and PGT135, which exhibited affinities in the low nanomolar range (Extended Data Fig. 6a). These neutralization experiments differ from the structural characterizations and SPR binding experiments, however, in that bivalent IgGs were used for neutralization assays, whereas monovalent Fabs were used for the electron microscopy (EM) and SPR analyses. Bivalent IgG binding could compensate for a weak Fab-antigen binding affinity; however, HIV-1 bNAb IgGs generally do not utilize avidity due to the relatively few Env trimers coating the virus and the positioning of conserved bNAb epitopes on Env, thus limiting inter- and intra-spike crosslinking, respectively[15]. To evaluate potential contributions of avidity in neutralization, molar neutralization ratios (MNRs) [$IC_{50}$ Fab (nM)/$IC_{50}$ IgG (nM)] can be calculated. We note that viruses with densely packed spikes can exhibit MNRs over 1,000, whereas MNRs for anti-HIV-1 bNAbs tend to be low[15]. Notable exceptions include V3 bNAbs PGT128 (ref. 30) and EPTC112 (ref. 13), which target similar Env epitopes as 007 (Fig. 2b, f and Extended Data Fig. 3b) and were reported to be ~30- to 2,000-fold more potent when formatted as an IgG than as a Fab against viruses tested. Although the mechanism underlying the enhanced IgG potencies was not reported, it was speculated to result from inter-spike crosslinking between adjacent Env trimers on the virion surface.

To investigate the possibility that 007 IgG utilizes avidity during neutralization of pseudovirions, we repeated in vitro neutralization assays to compare the neutralization potency of the monovalent 007 Fab to bivalent IgG1 and bivalent IgG3 forms of 007. To control for steric effects that may impact IgG neutralization due to the increased mass of an IgG compared to a Fab (~150 kDa for an IgG1 versus ~50 kDa for a Fab), we also created a bispecific 007 IgG1 in which one Fab arm was replaced with the anti-CD3 antibody OKT3[31], which does not recognize HIV-1 Env. We found bivalent IgG1 and bivalent IgG3 forms of 007 had

similar potencies, with a mean IgG1/IgG3 MNR of 0.82 across viruses tested (Fig. 3a, b and Supplementary Table 10), indicating that the longer IgG3 hinge does not confer improved neutralizing activity. Additionally, the monovalent 007 Fab and monovalent 007/OKT3 bispecific IgG1 had similar potencies, with a mean Fab/bispecific IgG1 MNR of 1.6 across viruses tested. However, against all viruses tested, the bivalent forms of 007 were more potent than monovalent forms, consistent with avidity effects that enhanced neutralization: bispecific IgG1/bivalent IgG1 MNRs ranged from 4.0 to 250 across the 26 viruses for which MNRs were derived (Fig. 3a, b and Supplementary Table 10). Such variation is expected since enhancements due to avidity depend on multiple factors including the dissociation rate of a Fab for an Env antigen[15], which is expected to vary for different viral strains. Notably, viruses that were less potently neutralized by monovalent forms of 007 benefitted more from antibody bivalency than viruses that were more potently neutralized by monovalent forms of 007 (Extended Data Fig. 6b, c).

A possible mechanism for IgG avidity effects is through intra-spike crosslinking (that is, both Fab arms on a bivalent IgG engage with epitopes on a trimeric Env), which has been previously inferred from Fab-trimer structures for other viral pathogens. For example, as described for anti-SARS-CoV-2 antibodies[32–34], a measured distance <~65 Å between the C termini of adjacent Fab heavy chains raises the possibility for intra-spike crosslinking by an IgG, as this would permit the C termini of two Fab heavy chains to come together to form the N terminus of the IgG Fc region. In our structure of BG505-DS SOSIP with two copies of 007 Fab, the measured distance between the C termini of adjacent Fab heavy chains, ~120 Å, was approximately twofold greater than this distance cutoff (Fig. 3c), suggesting that intra-spike crosslinking by 007 IgG interacting with the closed BG505 Env trimer in our structure would not be possible. However, one must also consider different structural states of the antigen that may be targeted. For example, double electron-electron resonance spectroscopy demonstrated that unliganded SOSIP Env trimers can sample open states that can be recognized by neutralizing antibodies, such as the occluded-open Env trimer state in which the gp120 protomers are outwardly rotated but the V1/V2 loop is not displaced to the sides of the Env trimer[35]. Docking the 007 Fab-gp120 coordinates onto the gp120s of an occluded-open Env placed the C termini of adjacent Fabs within 65 Å (Fig. 3c). Therefore, one possibility to account for the apparent involvement of avidity effects in 007 interactions with Env trimers is that 007 IgG may intra-spike crosslink to an occluded-open or other altered Env conformation not observed in our 007 Fab-SOSIP structures.

In the context of SARS-CoV-2 spike trimers, structural studies of IgGs interacting with stabilized trimers revealed intra-spike crosslinking[36,37]. Thus, in an attempt to investigate bivalent IgG interactions with Env trimer structurally, we incubated BG505 SOSIP with 007 IgG1 and imaged by cryo-EM (Extended Data Fig. 7 and Supplementary Table 8). Rather than observing intra-spike crosslinking, the most populated structural class of 007 IgG-BG505 complexes contained "trimer-dimers" in which two SOSIP Envs were crosslinked by Fabs from three IgG molecules (Fig. 3d). This assembly exhibited D3 symmetry, with the apexes of two Env trimers facing each other and separated by ~70 Å. Density for the Fc region of the IgG was not resolved, an expected consequence of flexibility at the IgG hinge region.

**Fig. 2 | 007 recognizes an $N332_{gp120}$ glycan-independent V3 epitope. a**, Left: overviews of the four structural classes identified of SOSIP trimers with 0-, 1-, 2- or 3-bound 007 Fabs per trimer. Right: the number of particles used in each of the final reconstructions. **b**, Overlay of 007 with V3-targeting bNAbs (PDB codes 5C7K, 5T3Z, 6CH7, 4JM2). **c**, Alignment of 007 VH and VL to their predicted germline V gene segments. 007 residues within 4 Å of protein or glycan components on Env are indicated by colored circles. **d**, Structure overview, highlighting proximal glycans. **e**, Protein contacts between 007 and Env. **f**, Protein contacts between EPTC112 and Env (PDB code 8C8T). **g**, EM density highlighting the $N156_{gp120}$ (left) and $N301_{gp120}$ (right) glycans. **h**, Neutralizing

activity ($IC_{50}$) of 007 and 10-1074 against HIV-1 pseudoviruses produced in the presence of kifunensine (kif). wt, wild type. **i**, Comparison of glycoform abundance between total BG505 SOSIP and 007-bound SOSIP at $N301_{gp120}$ determined by LC-MS/MS. Points represent the replicative measurements (three for total BG505 and two for bound BG505), and bar graphs represent the mean of replicative measurements. Differences between groups were evaluated for statistical significance based on $P$ values calculated using Welch's $t$-test (two-sided). The $P$ values are *0.04468 (for Man-6), **0.001095 (for Man-5) and ***0.04321 (for Man-4).

However, the distance between the C termini of the closest Fabs on the opposing Env trimers was 13 Å (Fig. 3d), consistent with these densities originating from a single IgG. In addition to the trimer-dimer structural class, trimer classes with 0, 1, or 2 copies of bound 007 IgG were also observed (Fig. 3d). In these structures, the unbound Fab and Fc of each IgG were unresolved, leading to EM densities closely matching the densities of the Fab-SOSIP complexes (Figs. 2a and 3d).

The observed trimer-dimer structure would be compatible with Envs attached to separate virions (Fig. 3d), thus raising the possibility that the 007 IgG neutralizes at least in part by aggregating virions.

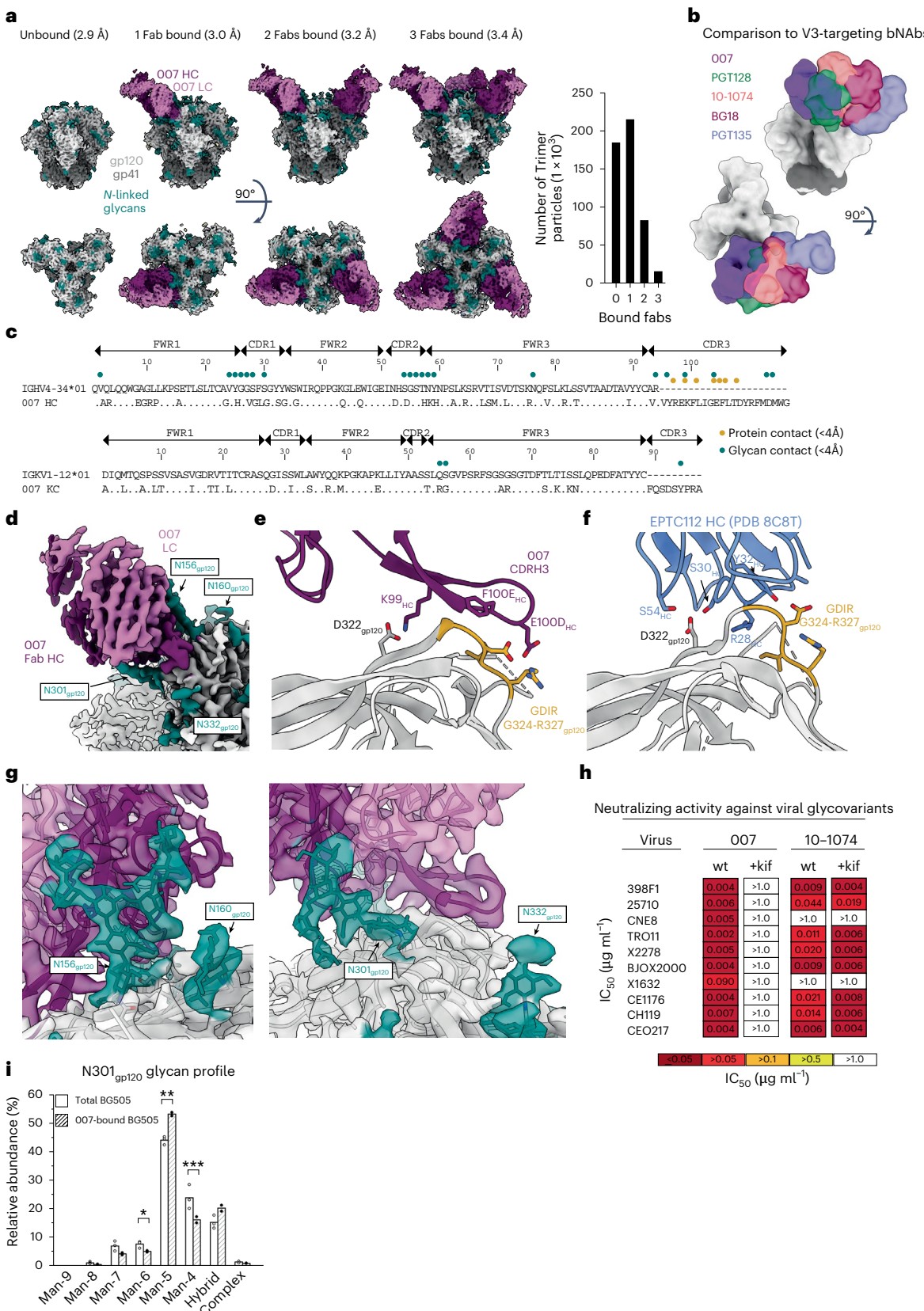

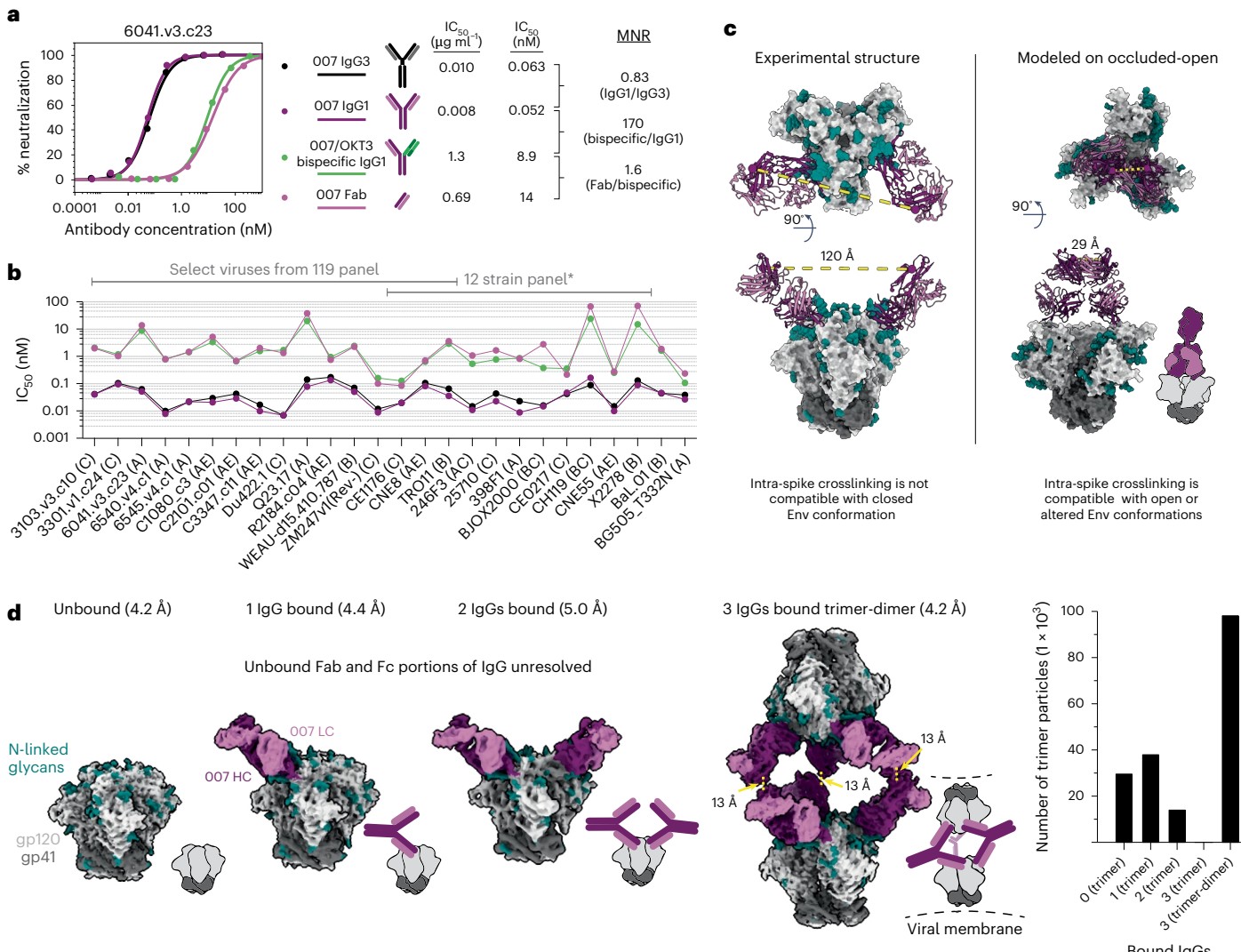

**Fig. 3 | Potent neutralizing activity requires 007 bivalency.** 007 exhibits avidity. **a**, Neutralization curves against strain 6041.v3.c23 (left) used to calculated MNRs comparing the IC$_{50}$ values for the IgG1, IgG3, bispecific IgG1 and Fab forms of 007 (right). **b**, Molar IC$_{50}$ values for IgG1, IgG3, bispecific IgG1 and Fab forms of 007 against a panel of HIV-1 strains. Clades are indicated in parentheses. *X1632 is not shown with the remainder of the 12-strain panel due to incomplete pseudovirus neutralization. IC$_{50}$ values and calculated MNRs are presented in Supplementary Table 10. **c**, Distance measurements between the C termini of the Fab heavy chains on the experimentally determined 2 Fab-bound trimer structure (left)

compared to a structure in which 007 Fab was modeled onto an occluded-open trimer conformation (PDB code 5VN8). A schematic showing a 007 IgG-bound occluded-open trimer is shown for clarity. **d**, Four structural classes identified after complexing 007 IgG with BG505 SOSIP (left). Schematics are included for clarity. Dashed line (yellow) indicates the distance measurements between C termini of Fab regions in 007 IgG-bound trimer-dimer structure. The number of particles used in each of the final reconstructions (right). The number of particles in the trimer-dimer class were multiplied by two, as each particle contained two SOSIP trimers.

However, the extent to which enhanced pseudovirus neutralization by 007 IgG can be attributed to viral aggregation, intra-spike crosslinking or another mechanism such as inter-spike crosslinking[13,30] or heteroligation[38] or by enhancing gp120 shedding[37] warrants further investigation.

## 007 exhibits high levels of antiviral activity and a complementary neutralization profile to canonical V3-specific bNAbs

To further characterize the neutralizing properties of bNAb 007 in comparison to other V3 loop-targeting bNAbs, we compared their activities against 107 common strains from the 119-strain multiclade pseudovirus panel consisting of difficult-to-neutralize (tier 2 and tier 3) viruses, major genetic HIV-1 subtypes and circulating recombinant forms (CRFs)[23] (Fig. 4a and Supplementary Table 7). Antibodies 10-1074 and PGT121 originate from the same donor and likely represent members of the same antibody lineage. Both were included to

provide an internal control for neutralization divergence within a single lineage. Notably, bNAb 007 demonstrated superior breadth and/or potency compared to reference V3 glycan site bNAbs across this panel (GeoMean IC$_{50}$/IC$_{80}$ = 0.01/0.03 µg ml⁻¹; breadth = 66%/55%). Moreover, 007 greatly exceeded the neutralizing activity of bNAb EPTC112[13], a recently identified antibody of the same epitope class. In addition, 007 demonstrated high neutralization activity against a 100-strain clade C panel[39] (GeoMean IC$_{50}$/IC$_{80}$ = 0.01/0.03 µg ml⁻¹; breadth = 68%/49%) (Extended Data Fig. 8a and Supplementary Table 11). Analyses of the antiviral activity against the 119-strain multiclade panel revealed that 007 displays a distinct neutralization profile compared to V3 glycan site reference bNAbs. Although reference V3 bNAbs only neutralized up to 75% of Clade AC (2 of 3 strains) and 67% of CRF AE viruses (10 of 15 strains), 007 achieved 100% and 80% breadth, respectively (Fig. 4b). Furthermore, unlike other V3 glycan site bNAbs, 007 maintained high neutralization coverage against viruses lacking the

$N332_{gp120}$ glycan (breadth= 71%) and those with amino acid substitutions in the $^{324}GD/NIR^{327}$ motif (breadth = 50%) (Fig. 4c). Consistent with structural analyses, the antiviral activity of 007 was dependent on the $N156_{gp120}$ and $N301_{gp120}$ glycans, with viruses lacking PNGSs at these positions being completely resistant to neutralization by 007 (Fig. 4c). These findings also align with the observed low to moderate correlation between V3 reference bNAbs and EN01 serum IgG neutralizing activity (Extended Data Fig. 1e). Beyond virus neutralizing activity, 007 also demonstrated strong Fc effector functions against Env expressing cells; in both FcγRIIIa signaling and natural killer cell-mediated antibody-dependent cytotoxicity (ADCC) assays (Extended Data Fig. 8b). Altogether, 007 exhibits a potent antiviral profile, via both neutralization and Fc effector function mechanisms.

Given its distinct neutralization profile and glycan dependency, we further analyzed the antiviral activity of 007 against 46 pseudovirus strains selected from the 119-strain multiclade pseudovirus panel[23]. This subset represents diverse HIV-1 clades and exhibits full resistance to the classical V3 glycan site bNAb 10-1074. Notably, although V3 reference bNAbs neutralized only 0% to 26% of these virus strains, 007 achieved a breadth of 65% with high potency (GeoMean $IC_{50}$ = 0.007 μg ml$^{-1}$) (Fig. 4d). Conversely, classical V3 glycan site bNAbs exhibited up to 62% neutralization breadth against a subset of 40 007-resistant pseudoviruses selected from the 119-strain multiclade panel, indicating functional complementarity (Fig. 4d). To further investigate how known Env escape mutations influence 007's neutralizing activity, we evaluated the sensitivity of HIV-1$_{BG505}$ and HIV-1$_{Tro11}$ pseudovirus variants[40]. Amino acid substitution at position $D325_{gp120}$ and removal of the PNGS at position $N332_{gp120}$ had variable effects on the activity of V3 reference bNAbs (Fig. 4e). Whereas the neutralizing activities of 10-1074 and BG18 were abrogated by all tested mutations at $N332_{gp120}$ and $S334_{gp120}$, only $N332D_{gp120}$ and $S334D_{gp120}$ greatly reduced PGT128 activity (Fig. 4e). Furthermore, among the V3 reference bNAbs tested, only 10-1074 was affected by $D325G_{gp120}$, whereas PGT121 remained relatively unaffected by all assessed mutations (Fig. 4e). In contrast, 007 neutralizing activity was impaired by amino acid substitutions at position $N156_{gp120}$ and $N301_{gp120}$, whereas it maintained high antiviral activity against typical V3 glycan site escape variants at positions $D325_{gp120}$, $N332_{gp120}$ and $S334_{gp120}$[40] (Fig. 4e). Supporting these findings, the combination of 007 with other V3 glycan site bNAbs complemented their individual neutralizing activity, leading to increased breadth and potency (>40-fold) of the combination against the global pseudovirus panel (Fig. 4f and Supplementary Table 12). In addition, in silico modeling of 007 and 10-1074 combination predicted complementary neutralizing activity, enhancing the breadth and potency against the 119 multiclade panel to 84.5% and a GeoMean $IC_{80}$ of 0.043 μg ml$^{-1}$ (Fig. 4g). The distinct neutralization profile of 007 establishes it as

complementary to V3 glycan site bNAbs, offering broader coverage of viruses that evade neutralization by antibodies of this class.

## Deep mutational scanning reveals a distinct escape profile

Deciphering viral escape pathways is essential to inform clinical applicability of bNAbs. To more comprehensively investigate viral escape from bNAb 007, we utilized a lentiviral pseudovirus-based deep mutational scanning (DMS) platform[41,42] using two HIV-1 Envs from distinct clades: TRO.11 (Clade B) and BF520.W14M.C2 (Clade A) (Extended Data Fig. 4a). This platform encompasses nearly all functionally tolerated mutations at each individual Env residue, enabling a systematic evaluation of their effects on bNAb sensitivity and viral cell entry in vitro. DMS revealed a distinct viral escape profile for 007 compared to classical V3-targeting bNAbs. Specifically, mutations at the $N332_{gp120}$ glycosylation site in both $Env_{TRO.11}$ and $Env_{BF520}$ conferred escape from classical V3-targeting bNAbs[10,11]. In contrast, 007 remained unaffected by these mutations (Fig. 5a and Extended Data Fig. 8c). Conversely, mutations that disrupted glycosylation at sites $N156_{gp120}$, $N188a_{gp120}$, and $N301_{gp120}$ in $Env_{TRO.11}$ reduced 007's neutralizing activity but did not impair the activity of most other V3-targeting bNAbs. The only exception was PGT128, which was similarly affected by $N301_{gp120}$ glycosylation loss (Fig. 5a). Due to functional intolerance of $Env_{BF520}$ to mutations at $N156_{gp120}$ and $N301_{gp120}$, mutations at these sites could not be reliably assessed for this virus strain (Extended Data Fig. 8c).

In addition to escape due to substitutions in PNGSs, DMS uncovered distinct escape pathways from 007 at non-glycosylated residues in the V3 loop. Although substitutions at $Env_{TRO.11}$ residue $R327_{gp120}$ facilitated escape from classical V3-targeting bNAbs, 007 was also affected by mutations at $Env_{TRO.11}$ residues $322-323_{gp120}$ and $Env_{BF520}$ residues $318-320_{gp120}$ (Fig. 5a and Extended Data Fig. 4a). Notably, substitutions introducing positive charges or eliminating negatively charged residues in this region ($D322_{TRO11}K/R$ or $D321_{BF520}K/H$) mediated escape from 007, whereas introducing a negative charge in this region in BF520 ($G322_{BF520}D/E$) enhanced neutralizing activity of 007, without impacting the neutralization capacity of other V3-targeting antibodies (Fig. 5a and Extended Data Fig. 8c), further highlighting the importance of the electrostatic interaction between $K99_{007}$ and $D322_{gp120}$ observed in the 007 structure with BG505.

To validate the viral escape pathways identified by DMS, we conducted TZM-bl neutralization assays incorporating the identified escape mutations in the HIV-1 $Env_{TRO.11}$ background. The results revealed a high concordance between escape profiles identified by DMS and changes in neutralization potency measured in TZM-bl neutralization assays (fold changes in $IC_{50}$s) (Fig. 5b and Supplementary Table 13). These assays confirmed the pattern of viral escape observed in DMS analyses. Amino acid substitutions in the V1, V2 and V3 loop, but not at the $N332_{gp120}$ PNGS, mediated escape from 007, setting this bNAb apart from classical V3-targeting antibodies.

---

**Fig. 4 | Distinct neutralization profile and $^{324}GD/NIR^{327}$ motif dependency of 007. a**, Neutralization breadth (%) and potency ($IC_{50}/IC_{80}$) of 007 against the 119 multiclade pseudovirus panel[23]. Curve graphs illustrate the breadth as a function of $IC_{50}$ (top) and $IC_{80}$ (bottom). **b**, Illustration of the neutralization profile of 007 in comparison to V3 glycan site reference bNAbs against different virus clades of the 119 multiclade panel. Breadth against different HIV-1 clades is illustrated as radar plot (top) and summarized in tabular form (bottom). **c**, Dependency of 007 and V3 glycan site reference bNAbs on potential N-linked glycosylation sites and the $^{324}GD/NIR^{327}$ motif. Breadth against different HIV-1 variants with differing in glycosylation patterns and $^{324}GD/NIR^{327}$ motif sequence is illustrated as radar plot (top) and summarized in tabular form (bottom). Subst., amino acid substitution. **d**, Complementary neutralizing activity (GeoMean $IC_{50}$, breadth) of 007 and classical V3 glycan site bNAbs against panels of resistant pseudovirus strains. Pie charts illustrate the clade distribution of resistant pseudovirus strains. **e**, Neutralizing activity against viral escape mutations located at 007 contact residues (N156 and N301) and within the V3 glycan site of the HIV-1 Env trimer. The top row displays bNAb $IC_{50}$ values for the $BG505_{T332N}$ and/or Tro11

pseudovirus, while panels illustrate changes in bNAb sensitivity ($IC_{50}$ fold change) of virus mutants relative to the wild-type pseudovirus strain. Antibodies were tested in duplicates. Asteriks indicate mutants where the $IC_{50}$ fold changes were determined in the Tro11 backbone. **f**, Neutralizing activity of 007 in combination with V3 glycan site bNAbs (mixed at a 1:1 ratio) against the global pseudovirus panel[18]. Single and combined mAbs were tested up to a concentration of 1 μg ml$^{-1}$ (total IgG amount). Red numbers indicate the fold change in $IC_{50}$s (increase in potency) between the individual mAb and its combination with 007. **g**, Computational modeling of the predicted neutralizing activity of bNAb 007 in combination with 10-1074 against the 119 multiclade pseudovirus panel[23] using the CombiNaber tool[50] (http://www.hiv.lanl.gov/content/sequence/COMBINABER/combinaber.html). Breadth (%) was calculated using a cutoff of ≤10 μg ml$^{-1}$ (**a**–**d**,**g**). Data are shown for identical virus strains across each panel with available reference neutralization data (**a**–**d**,**g**). Reference bNAb data were sourced from the CATNAP database. 10-1074 and PGT121 derive from the same donor and likely share a common lineage.

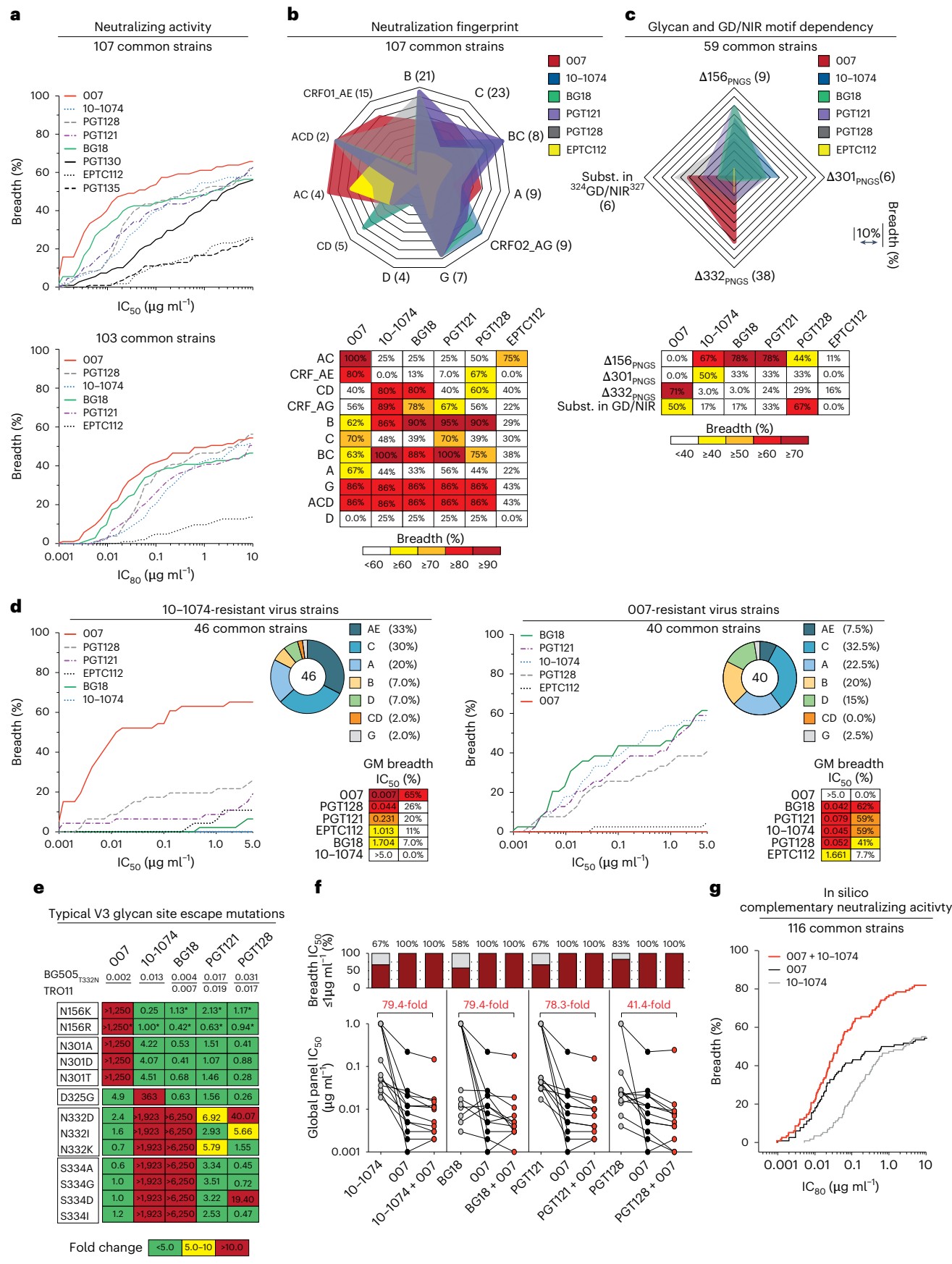

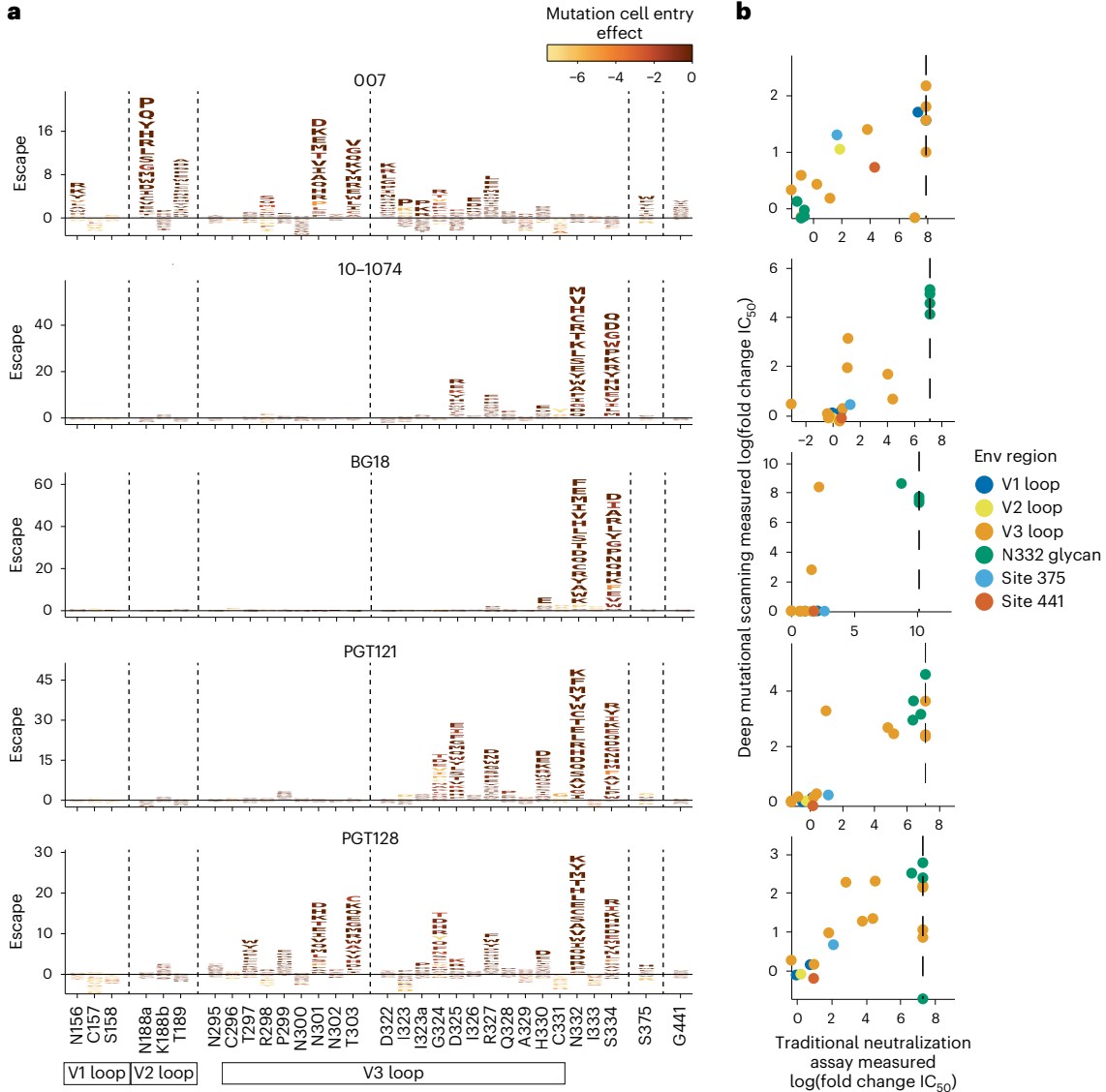

**Fig. 5 | Deep mutational scanning analyses reveal distinct viral escape from 007. a**, Logo plots showing effects of mutations on neutralization escape in HIV Env$_{TRO.11}$ for antibodies 007, 10-1074, BG18, PGT121 and PGT128. The height of each letter represents the effect of that amino acid mutation on antibody neutralization, with positive heights (letters above the zero line) indicating mutations that cause escape, and negative heights (letters below the zero line) indicating mutations that increase neutralization. Letters are colored by the effect of that mutation on Env-mediated cell entry function, with yellow corresponding to reduced cell entry and brown corresponding to neutral effects on cell entry. Only key sites are shown. See https://dms-vep.org/HIV_Envelope_TRO11_DMS_007/htmls/all_antibodies_and_cell_entry_overlaid.html for interactive versions of the escape maps that show all mutations. Escape

maps against HIV-1 Env$_{BF520}$ are shown in Extended Data Fig. 8c. **b**, Scatter plots of Env$_{TRO.11}$ mutant fold-change IC$_{50}$s measured by deep mutational scanning versus those measured in traditional neutralization assays. Each scatter plot shows log fold change IC$_{50}$s for neutralization assays using the antibody labeling the logo plot in the same row. Each point represents the mean of two replicate neutralization curve measurements of one Env$_{TRO.11}$ mutant. Env$_{TRO.11}$ mutants are colored by Env region or site. Vertical dotted lines represent the limit of detection of the neutralization assays. See Methods, "Deep mutational scanning data analysis", for details on how deep mutational scanning measured escape values are converted to deep mutational scanning measured IC$_{50}$ values. The deep mutational scanning measured effects of mutations on escape from antibody 10-1074 shown in this figure were previously published[41].

## 007 suppresses viremia in vivo, and dual targeting of the V3 glycan site affects viral rebound

To investigate the in vivo activity of 007 and evaluate viral escape profiles, we investigated its effects in humanized mice infected with HIV-1$_{ADA}$ (NOD.Cg-Rag$^{1tm1mom}$Il2rg$^{tm1Wjl}$/SzJ (NRG) mice; $n = 33$; 19 females, 14 males, aged 6–8 months). Following a loading dose of 1 mg IgG, infected mice were treated with 0.5 mg of either 007 or 10-1074 IgG twice a week for up to 5 weeks (Fig. 6a and Extended Data Fig. 9). PBS-treated mice with matched stem cell donors served as controls, and log$_{10}$ changes in viremia were normalized to this control group to account for declining viral loads in the PBS-treated animals (Extended Data Fig. 9).

Similar to 10-1074 IgG, administration of 007 IgG monotherapy resulted in a rapid decline in viral loads by 0.78 log$_{10}$ copies ml$^{-1}$, followed by rebound of viremia within 14 days after treatment initiation (Fig. 6a). To characterize viral escape associated with rebound, we performed single-genome sequencing (SGS) of plasma virus collected 5 weeks after treatment initiation and determined sensitivities to the corresponding bNAb (Fig. 6b and Supplementary Table 14). Viral escape from 10-1074 monotherapy was associated with mutations that abrogated the PNGS at N332$_{gp120}$. In contrast, 13 env sequences from four mice receiving 007 monotherapy exhibited mutations at residues 139$_{gp120}$, 146$_{gp120}$, 303$_{gp120}$, 322$_{gp120}$ and 341$_{gp120}$ in the V1/V2 or V3 loop. Of these,

only mutations at residues $303_{gp120}$ and $322_{gp120}$ mediated resistance to 007. Notably, all generated escape variants from 10-1074 monotherapy retained in vitro sensitivity to 007 and vice versa (Fig. 6b and Supplementary Table 14). We conclude that bNAb 007 reduces viremia in vivo and exerts a distinct and less restricted selection pressure on HIV-1 compared to 10-1074.

The distinct viral escape profile of 007 led us to explore whether dual targeting the conserved V3 glycan site could force the virus to accumulate mutations in this conserved epitope, potentially reducing viral fitness and prolonging viral suppression. To investigate this, we sequentially administered 007 IgG to animals pretreated with 10-1074 (and vice versa) once their viral loads rebounded to baseline levels due to the emergence of resistant escape variants (1 mg loading dose followed by 0.5 mg twice weekly for each antibody) (Fig. 6c). To maintain selection pressure, the initial antibody therapy (10-1074 or 007) was continued while the complementary antibody was added. Despite circulating viral variants resistant to the initially applied antibody, sequential addition of 007 to 10-1074 treatment (and vice versa) led to a transient suppression of viremia in all treated animals, indicating that 007 can overcome 10-1074 escape in vivo and vice versa (Fig. 6c). Next, we examined whether initiating treatment with a combination of 007 and 10-1074 IgGs would exert greater selection pressure and demonstrate enhanced in vivo efficacy compared to monotherapy or sequential combination therapy (Fig. 6d). Administration of combination therapy resulted in prolonged suppression of viremia (20 days versus 30 days average time to viral rebound, defined as ≥0.5 $log_{10}$ viral load increase relative to nadir) demonstrating in vivo synergy (Fig. 6d). SGS of plasma rebound viruses from the sequential and initial combination therapy group revealed selection of combined escape mutations in the V1/V2 and V3 loop residues previously identified from the antibody monotherapy groups ($N332_{gp120}$, $146_{gp120}$, $303_{gp120}$ or $322_{gp120}$) (Fig. 6e, f and Supplementary Table 15). Our findings demonstrate that antibody 007 can overcome 10-1074-class viral escape in vivo and that a V3 dual-targeting strategy enhances in vivo efficacy.

## Discussion

Advances in donor screening and antibody isolation techniques have accelerated the discovery of potent, broadly neutralizing anti-HIV-1 antibodies[1,3]. Clinical trials have demonstrated their promise for prevention and treatment of HIV-1 infections and informed vaccine design efforts[1,3]. However, similar to monotherapy with antiretroviral agents, administration of single bNAbs results in the rapid selection of resistant viral variants and fails to achieve the requisite antiviral activity for clinical success[2,40,43]. Application of bNAb combinations with complementary neutralization coverage have demonstrated enhanced neutralizing activity and long-term control of viremia in preclinical and clinical settings[2]. Similarly, the development of a fully protective vaccine will likely require the induction of bNAbs targeting multiple HIV-1 Env epitopes, emphasizing the need for the identification and characterization of bNAbs directed against novel antigenic sites.

In this study, we characterized the anti-HIV-1 bNAb 007 targeting the V3 glycan site of the Env trimer through a distinct binding mode. The V3 glycan site is centered around the PNGS at residue $N332_{gp120}$ and extends to high-mannose and complex-type N-glycans at positions $N156_{gp120}$, $N295_{gp120}$, $N301_{gp120}$, $N339_{gp120}$, $N386_{gp120}$ and $N392_{gp120}$ as well as the underlying protein surface[9]. bNAb lineages directed against the V3 glycan site recognize the $^{324}$GD/NIR$^{327}$ protein motif, exhibiting variability in N-glycan accommodation and binding angles of approach[10,11,30]. Unlike classical V3 glycan site bNAbs[10,11,40], 007 does not require the $N332_{gp120}$ glycan for binding. Instead, it interacts primarily with glycans at positions $N156_{gp120}$ and $N301_{gp120}$. In terms of glycan dependency, 007 shares similarities with the recently identified nAb EPTC112[13]. However, 007 differs in its CDRH3-mediated mode of binding, more pronounced interactions with the $^{324}$GD/NIR$^{327}$ protein motif, and, most importantly, its superior antiviral activity (66% versus 28% breadth; GeoMean $IC_{50}$ = 0.01 μg ml$^{-1}$ versus 0.32 μg ml$^{-1}$; cutoff ≤10 μg ml$^{-1}$; 107 strains). The enhanced neutralizing activity of 007 is an important consideration for the potential utility of this epitope for vaccine design, therapy and/or prevention.

The V3 glycan site represents a prime target for vaccine design[3,44], as bNAb responses directed against the $N332_{gp120}$-supersite are among the most frequently elicited in individuals infected with HIV-1 (ref. 45). Moreover, the diverse binding poses, varying glycan dependencies, distinct antibody gene usage and the moderate level of somatic hypermutation observed in some V3-targeting bNAbs further underscore the potential of the V3 glycan site as a favorable target for immunogen design[44,46]. Whereas current immunization strategies seek to elicit $N332_{gp120}$ glycan-specific antibody responses[3,44,46], the $N332_{gp120}$-independent V3 epitope has not been considered. In this context, 007 represents a promising avenue for vaccine development due to its enhanced antiviral activity and sequence features. Similar to the V3 bNAb BG18, a target of vaccine design[46], 007 exhibits a shorter CDRH3 compared to other characterized V3-targeting bNAbs and lacks insertions or deletions in both its heavy and light chains[46]. These features may enhance the feasibility of eliciting 007-like bNAbs by vaccination. However, the high degree of somatic hypermutation in

**Fig. 6 | 007 monotherapy and dual V3 glycan site targeting in vivo.**
**a**, Investigation of the antiviral activity of 10-1074 and 007 monotherapy in HIV-1$_{ADA}$-infected humanized mice (NOD.Cg-Rag$^{1tm1mom}$Il2rg$^{tm1Wjl}$/SzJ (NRG) mice). Graphs display the absolute HIV-1 RNA plasma copies ml$^{-1}$ (top) and relative $log_{10}$ changes from baseline viral loads (bottom) after initiation of bNAb therapy. $Log_{10}$ changes were normalized to the viral loads observed in the PBS control group (Extended Data Fig. 9). Dashed lines (top graphs) indicate the lower limit of quantitation (LLQ) of the qPCR assay (260 copies ml$^{-1}$). Green lines indicate mean viral loads ± s.d. in the PBS control group (n = 5), and red lines display the adjusted average $log_{10}$ changes from baseline (day −2), corrected for changes in the control group. **b**, Analyses of single HIV-1 plasma *env* sequences from HIV-1$_{ADA}$-infected humanized mice obtained after viral rebound on day 35 for 007 and 10-1074 monotherapy groups. Total number of analyzed sequences is indicated in the center of each pie chart. Mice are labeled according to icon legends in panel **a**. Colored bars on the outside of the pie charts indicate mutations in V1/V2 loop and V3 loop. Sensitivity ($IC_{50}$s) of pseudoviruses generated from SGS-derived sequences against 007 and 10-1074. **c**, Sequential treatment with 007 or 10-1074 in HIV-1$_{ADA}$-infected humanized mice following viral rebound during 007 or 10-1074 monotherapy (from **a**). This approach included maintaining 007 or 10-1074 monotherapy while integrating 007 or 10-1074 in the treatment regimen. Dashed lines (top graphs) indicate the LLQ of the qPCR assay (260 copies ml$^{-1}$). Green lines indicate mean viral loads ± s.d. in the PBS control group (n = 3), and red lines display the adjusted average $log_{10}$ changes from baseline (day 35), corrected for changes in the control group. **d**, Antiviral activity of 007 and 10-1074 combination therapy in HIV-1$_{ADA}$-infected humanized mice. Graphs display the absolute HIV-1 RNA plasma copies ml$^{-1}$ (top) and relative $log_{10}$ changes from baseline viral loads (bottom) after initiation of bNAb therapy. Dashed lines (top graphs) indicate the LLQ of the qPCR assay (260 copies ml$^{-1}$). Green lines indicate mean viral loads ± s.d. in the PBS control group (n = 5), and red lines display the adjusted average $log_{10}$ changes from baseline (day −2), corrected for changes in the control group. **e**, Analyses of single HIV-1 plasma *env* sequences from HIV-1$_{ADA}$-infected humanized mice obtained after viral rebound for 007 and 10-1074 combination therapy groups. Total number of analyzed sequences is indicated in the center of each pie chart. Mice are labeled according to icon legends in **a**. Colored bars on the outside of the pie charts indicate mutations in V1/V2 loop and V3 loop. Sensitivity ($IC_{50}$s) of pseudoviruses generated from SGS-derived sequences against 007 and 10-1074. **f**, Alignment of plasma SGS-derived *env* sequences obtained from individual mice (*y* axis) after viral rebound following mono- or sequential therapy. *Env* sequences are shown as horizontal gray bars from residues 100 to 500 relative to HXB2 (*x* axis). Mutations identified from day −2 are indicated in black, mutations after viral rebound following monotherapy in red (**a** and **b**), and following administration of 007 and 10-1074, either sequentially or in combination in blue (**c**, **d** and **e**). Sites at which selected mutations can confer resistance are highlighted by vertical blue bars.

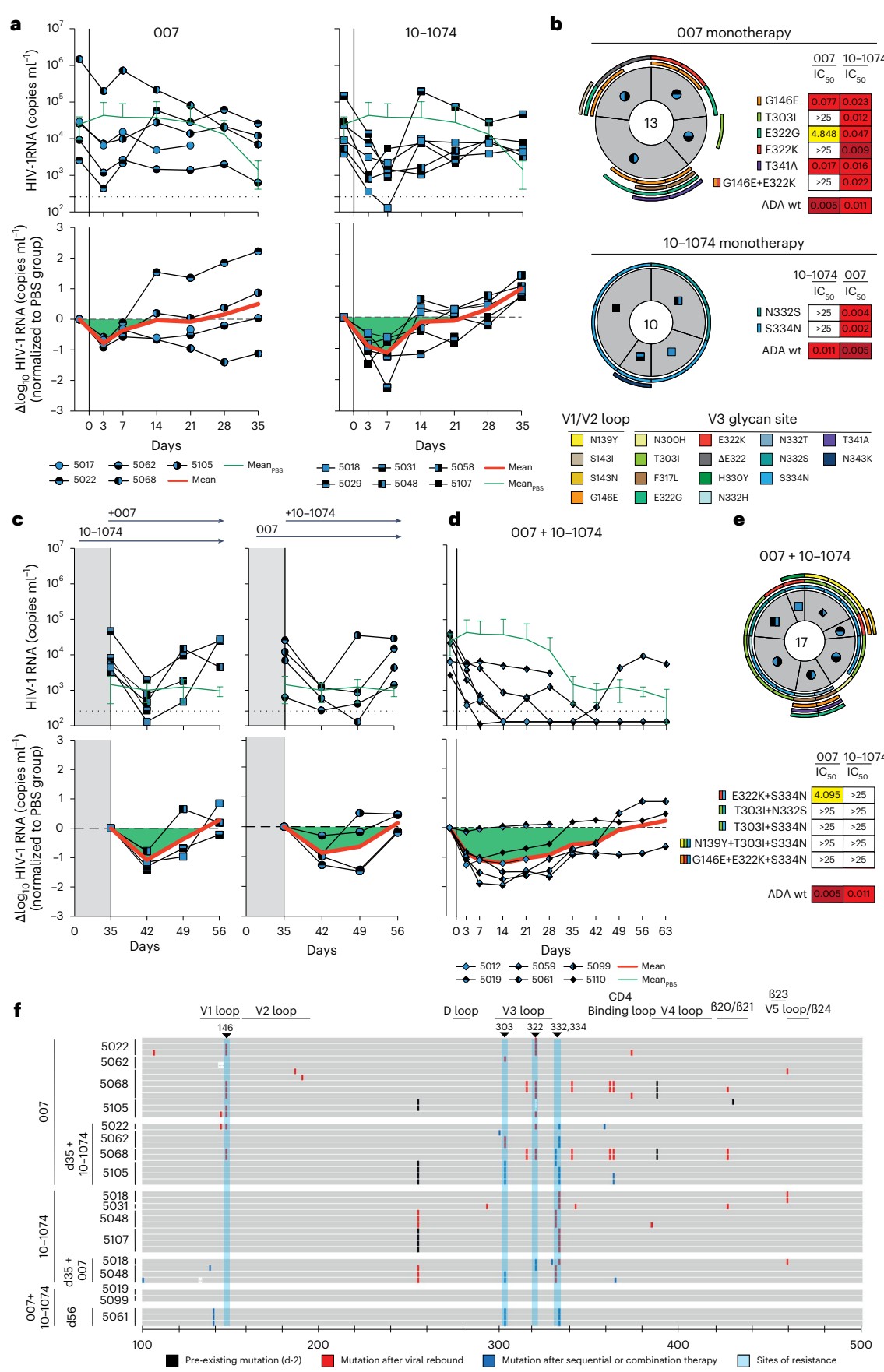

007 could pose a challenge for inducing similar bNAbs via vaccination, necessitating germline reversion studies to determine whether minimally mutated variants retain antiviral activity[46].

Due to its distinct epitope and neutralization profile, 007 represents a promising potential candidate for therapy, functional cure, and prevention strategies involving passive administration of bNAb combinations. This is of particular interest in regions of the world with a high prevalence of CRF01AE viruses that represent a coverage gap of V3 glycan site bNAbs due to the lack of the $N332_{gp120}$ glycan[47]. Indeed, 007 retains high levels of antiviral activity against CRF AE and AC viruses and could efficiently substitute classical V3 glycan site bNAbs in combination regimens considered for these regions. The distinct V3-binding mode and escape profile of 007 have important clinical implications, as they contribute to 007's potential to effectively complement other V3 glycan site bNAbs, which, despite their high potency, often exhibit limited breadth[10,11]. The ability of 007 to overcome 10-1074 escape both in vitro and in vivo, along with its predicted combined breadth of 84.5% in combination with 10-1074, highlight new opportunities for bNAb combination strategies, including dual targeting of the V3 glycan site to enhance breadth, potency and selection pressure on this conserved epitope. With regard to accelerated clearance of 007 in huFcRn mice compared to clinically evaluated bNAbs, future pharmacokinetic analyses will be important to optimize dosing strategies for clinical application.

Structural analysis of the 007 Fab in complex with a SOSIP Env trimer revealed sub-stoichiometric Fab binding, a surprising finding for a potent bNAb. Interestingly, EPTC112, which recognizes a similar epitope, also lacked full Fab occupancy in its structure, with only two Fabs per trimer modeled[13]. Similar to results with EPTC112[13], in vitro neutralization assays for 007 demonstrated IgG bivalency enhanced neutralization potencies. Based on our 007 Fab-SOSIP structures, we hypothesized that intra-spike bivalent IgG binding could occur on open forms of an Env trimer. However, when SOSIP-IgG complexes were analyzed by cryo-EM, a trimer-dimer of closed Env-IgG complexes was observed, a configuration that could occur if IgGs link Envs trimers on two separate virions. Analogous structures linking two SARS-CoV-2 spike trimers by IgGs were postulated to enhance neutralization by aggregating virions[14], and cryo-electron tomography of nAbs incubated with SARS-CoV-2 demonstrated inter-virion bivalent binding of IgGs[48]. The SOSIP Env trimers we used for structure determinations contained specific mutations ($I559P_{gp41}$, $A501C_{gp120}$ $T605C_{gp41}$) that decrease the sampling of alternative Env conformations, whereas neutralization assays use membrane-bound Env trimers lacking these mutations. Further studies of 007 IgG in complex with HIV-1 virions or virus-like particles displaying native Env trimers will be required to elucidate its mechanism of bivalent neutralization. In summary, the favorable neutralization properties and distinct viral escape profile position 007 as a promising investigational candidate for HIV-1 therapy, functional cure, and prevention. Our findings further emphasize the $N332_{gp120}$ glycan-independent V3 epitope as a compelling target for vaccine development.

## Online content

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

Lutz Gieselmann[1,2,25], Andrew T. DeLaitsch [3,25], Malena Rohde[1,25], Caelan Radford[4], Johanna Worczinski[1], Anna Ashurov[1], Elvin Ahmadov[1], Judith A. Burger[5,6], Colin Havenar-Daughton [7], Sharvari Deshpande[7], Federico Giovannoni[8], Davide Corti [8], Christoph Kreer [1], Meryem Seda Ercanoglu[1], Philipp Schommers [1,2,9,10], Ivelin S. Georgiev[11,12,13,14,15], Anthony P. WestJr. [3], Jacqueline Knüfer[1], Ricarda Stumpf[1], Arne Kroidl [16,17], Christof Geldmacher[16,17], Lucas Maganga[18], Wiston William[18], Nyanda E. Ntinginya[18], Michael Hoelscher [16,17,19,20], Zhengrong Yang[21], Qing Wei[22], Matthew B. Renfrow[21], Todd J. Green[22], Jan Novak [22], Marit J. van Gils [5,6], Harry B. Gristick[3], Henning Gruell [1], Jesse D. Bloom [4,23], Michael S. Seaman[24], Pamela J. Bjorkman [3] & Florian Klein [1,2,9] ✉

[1]Laboratory of Experimental Immunology, Institute of Virology, Faculty of Medicine and University Hospital Cologne, University of Cologne, Cologne, Germany. [2]German Center for Infection Research, Partner Site Bonn-Cologne, Cologne, Germany. [3]Divison of Biology and Biological Engineering, California Institute of Technology, Pasadena, CA, USA. [4]Basic Sciences Division and Computational Biology Program, Fred Hutchinson Cancer Center, Seattle, WA, USA. [5]Department of Medical Microbiology and Infection Prevention, Laboratory of Experimental Virology, Amsterdam UMC, University of Amsterdam, Amsterdam, Netherlands. [6]Amsterdam Institute for Immunology and Infectious diseases, Amsterdam, Netherlands. [7]Vir Biotechnology, San Francisco, CA, USA. [8]Vir Biotechnology, Bellinzona, Switzerland. [9]Center for Molecular Medicine Cologne (CMMC), University of Cologne, Cologne,

Germany. [10]Department I of Internal Medicine, Faculty of Medicine and University Hospital Cologne, Cologne, Germany. [11]Vanderbilt Center for Antibody Therapeutics, Vanderbilt University Medical Center, Nashville, TN, USA. [12]Department of Pathology, Microbiology and Immunology, Vanderbilt University Medical Center, Nashville, TN, USA. [13]Vanderbilt Institute for Infection, Immunology and Inflammation, Vanderbilt University Medical Center, Nashville, TN, USA. [14]Department of Computer Science, Vanderbilt University, Nashville, TN, USA. [15]Center for Structural Biology, Vanderbilt University, Nashville, TN, USA. [16]Institute of Infectious Diseases and Tropical Medicine, LMU University Hospital, LMU Munich, Munich, Germany. [17]German Center for Infection Research (DZIF), Partner Site, Munich, Germany. [18]Mbeya Medical Research Centre, National Institute for Medical Research, Mbeya, Tanzania. [19]Fraunhofer Institute for Translational Medicine and Pharmacology ITMP, Immunology, Infection and Pandemic Research, Munich, Germany. [20]Unit Global Health, Helmholtz Zentrum München, German Research Center for Environmental Health (HMGU), Neuherberg, Germany. [21]Department of Biochemistry and Molecular Genetics, University of Alabama at Birmingham, Birmingham, AL, USA. [22]Department of Microbiology, University of Alabama at Birmingham, Birmingham, AL, USA. [23]Howard Hughes Medical Institute, Chevy Chase, MD, USA. [24]Center for Virology and Vaccine Research, Beth Israel Deaconess Medical Center, Harvard Medical School, Boston, MA, USA. [25]These authors contributed equally: Lutz Gieselmann, Andrew T. DeLaitsch, Malena Rohde. ✉e-mail: florian.klein@uk-koeln.de

## Methods

### Study participants and collection of clinical samples

Large blood draws and leukapheresis samples were collected in accordance with protocols reviewed and approved by the Institutional Review Board of the University of Cologne (study protocols 13-364 and 16-054) and local institutional review boards. Study participants were recruited from private practices and/or hospitals in Germany (Cologne, Essen and Frankfurt), Cameroon (Yaoundé), Nepal (Kathmandu) and Tanzania (Mbeya), and all participants provided written informed consent. Compensation was provided in line with institutional and ethical guidelines to reimburse time and expenses without exerting undue influence. A total of 2,354 serum samples were screened for anti-HIV-1 neutralizing activity to identify HIV-1 elite neutralizers[17]. Study individual EN01 was selected for a large blood draw and subsequent B cell isolation. Biosample collection was conducted irrespective of sex/gender, which was not a study design criterion. Clinical information was obtained from medical records.

### Cell lines

HEK293T cells (American Type Culture Collection, #CRL-3216) were cultured in DMEM (Thermo Fisher Scientific) supplemented with 10% fetal bovine serum (FBS; Sigma-Aldrich), 1x antibiotic-antimycotic (Thermo Fisher Scientific), 1 mM sodium pyruvate (Gibco) and 2 mM L-glutamine (Gibco) at 37 °C in an atmosphere containing 5% $CO_2$. HEK293-6E cells (National Research Council of Canada, Cat. No. NR16-179/2017E) were grown in FreeStyle 293 Expression Medium (Life Technologies) supplemented with 0.2% penicillin/streptomycin and maintained under constant agitation at 90 to 120 rpm at 37 °C and 6% $CO_2$. TZM-bl cells (NIH AIDS Reagent Program, #ARP5011) were cultured in DMEM supplemented with 10% FBS, 1 mM sodium pyruvate, 2 mM L-glutamine 50 µg ml$^{-1}$ gentamicin (Merck), and 25 mM HEPES (Millipore) at 37 °C in 5% CO2. All three cell lines (HEK293T, HEK293-6E and TZM-bl) were of female origin and were not specifically authenticated.

### Mouse models

NOD.Cg-Rag$^{1tm1mom}$Il2rg$^{tm1Wjl}$/SzJ (NRG) mice ($n$ = 33; 19 females, 14 males, aged 6–8 months) were acquired from The Jackson Laboratory and subsequently bred and housed within the Decentralized Animal Husbandry Network (Dezentrales Tierhaltungsnetzwerk) at the University of Cologne. Mice were maintained under specific pathogen-free conditions with a 12-h light/dark cycle at 20–22 °C and 30% to 60% humidity. Breeding mice were provided ssniff 1124 breeding feed, whereas experimental mice received ssniff 1543 maintenance feed. The generation of humanized mice followed an established protocol, with slight modifications[51,52]. Human CD34$^+$ hematopoietic stem cells were isolated from umbilical cord blood and placental tissue through immunomagnetic separation using CD34 microbeads (Miltenyi Biotec). The collection of these tissue sources was conducted with prior written informed consent, following protocols approved by the Institutional Review Board of the University of Cologne (16-110) and the Ethics Committee of the Medical Association of North Rhine (2018382). Within 5 days after birth, NRG mice received sublethal irradiation, after which human CD34$^+$ stem cells were administered via intrahepatic injection 4 to 6 h later. The engraftment and humanization efficiency were assessed 12 weeks after injection using FACS to detect peripheral blood mononuclear cells (PBMCs) in circulation[51]. All animal experiments were performed in compliance with ethical regulations and approved by the State Agency for Nature, Environmental Protection, and Consumer Protection of North Rhine-Westphalia (LANUV).

### PBMCs and plasma isolation

PBMCs were isolated from large-volume blood samples using density gradient centrifugation with Histopaque separation medium (Sigma-Aldrich) and Leucosep cell tubes (Greiner Bio-One), following the manufacturer's instructions. The purified PBMCs were cryopreserved at −150 °C in a freezing medium composed of 90% FBS and 10% dimethylsulfoxide until further use. Plasma samples were collected separately and stored at −80 °C for subsequent analyses.

### Isolation of single HIV-1-reactive B cells

The isolation of single antigen-reactive B cells was carried out following previously established methods[21,53]. CD19$^+$ B cells were selectively enriched from PBMCs using immunomagnetic separation with CD19 microbeads (Miltenyi Biotec) according to the manufacturer's protocol. Isolated CD19$^+$ B cells were subsequently stained on ice for 20 min with 4′,6-diamidino-2-phenylindole (DAPI; Thermo Fisher Scientific), anti-human CD20-Alexa Fluor 700 (BD), anti-human IgG-APC (BD) and GFP-labeled BG505$_{SOSIP.664}$[54]. Following staining, DAPI$^-$ CD20$^+$ HIV-1 Env-reactive IgG$^+$ single cells were sorted into 96-well plates using a FACSAria Fusion cell sorter (Becton Dickinson). Each well was preloaded with 4 µl of sorting buffer containing 0.5× PBS, 0.5 U µl$^{-1}$ RNAsin (Promega), 0.5 U µl$^{-1}$ RNaseOut (Thermo Fisher Scientific), and 10 mM dithiothreitol (DTT; Thermo Fisher Scientific). The plates were immediately cryopreserved at -80 °C following cell sorting.

### Amplification and analysis of heavy and light chain V gene sequences

Antibody heavy and light chain amplification from single cells was primarily conducted as described in prior studies[21,55,56]. Reverse transcription was carried out using Random Hexamers (Invitrogen) and Superscript IV (Thermo Fisher Scientific) in the presence of RNase inhibitors RNaseOUT (Thermo Fisher Scientific) and RNasin (Promega) to preserve RNA integrity. The synthesized cDNA was subsequently used for the amplification of immunoglobulin heavy and light chains using PlatinumTaq HotStart polymerase (Thermo Fisher Scientific), supplemented with 6% KB extender and gene-specific primer mixes targeting V gene regions. A semi-nested PCR strategy was applied with optimized V gene-specific primer mixes[57] to improve amplification efficiency[21,55–57]. The resulting PCR products were evaluated via gel electrophoresis to confirm expected fragment sizes before undergoing Sanger sequencing. Raw sequence processing, annotation and clonal assignment was performed with the Antibody Repertoire Toolkit (AbRAT)[58] using default settings. In brief, chromatograms were filtered to only retain sequences with a mean Phred quality score of at least 28 and a minimum read length of 240 nt. Variable region annotation, spanning from FWR1 to the end of the J gene segment, were annotated based on IgBLAST[59]. Nucleotide positions within the variable region exhibiting Phred scores below 16 were masked, and sequences containing more than 15 masked bases, premature stop codons or frameshifts were excluded from downstream analyses.

For clonal lineage assignment, productive heavy chain sequences were grouped by identical $V_H$/$J_H$ gene usage and clustered with AbRATs "Iterative Greedy CDR3 Clustering"-algorithm that is based on the pairwise Levenshtein distance between CDRH3s. Clonal clusters were defined by initiating the grouping process from a randomly selected sequence, with membership requiring a minimum of 75% CDRH3 amino acid identity (relative to the shortest sequence). To enhance classification accuracy, 100 iterations of randomization and clonal assignment were performed, with the configuration yielding the lowest number of unassigned sequences chosen for subsequent analysis. All assigned clone groups were manually validated by investigators, incorporating shared somatic mutations and light chain pairing information to ensure consistency in lineage identification.

### Cloning and production of monoclonal antibodies

The heavy and light chain V gene regions of mAbs were synthesized as eBlocks gene fragments (IDT), incorporating complementary overhangs pre-configured for cloning into expression vectors (IgG1, AbVec2.0-IGHG1, Addgene accession #80795; IgG3, AbVec2.0-IGHG3, Addgene accession #99577; Igλ, AbVec1.1-IGLC2-XhoI, Addgene

accession #99575; Igκ, AbVec1.1-IGKC, Addgene accession #80796). Cloning was performed using sequence- and ligation-independent cloning with T4 DNA polymerase (New England Biolabs) and chemically competent *Escherichia coli* DH5α, following established protocols[21,53,55,56,60]. Positive transformants were identified through Sanger sequencing, after which confirmed bacterial colonies were expanded in LB medium. Plasmid DNA was subsequently extracted using the NucleoBond Xtra Midi kit (Macherey-Nagel).

MAbs were generated by co-transfecting HEK293-6E cells ($0.8 \times 10^6$ cells in 50 ml) with heavy chain (IgG1) and light chain (Igλ or Igκ) expression plasmids using PEI (Sigma-Aldrich) as the transfection reagent. Cells were maintained at 37 °C with 5% $CO_2$ in FreeStyle 293 Expression Medium (Thermo Fisher Scientific) supplemented with 0.2% penicillin/streptomycin under continuous agitation at 120 rpm. Culture supernatants were harvested 7 days after transfection via centrifugation and incubated overnight at 4 °C with Protein G-coupled beads (GE Life Sciences). The beads were subsequently transferred to chromatography columns (Bio-Rad), washed with DPBS (Thermo Fisher Scientific), and antibodies were eluted using 0.1 M glycine (pH 3.0) into 1 M Tris (pH 8.0) to neutralize acidity. A final buffer exchange to PBS was performed using 30 K Amicon spin membranes (Merck Millipore). Antibody concentrations were determined via UV spectrophotometry using a Nanodrop system (Thermo Fisher Scientific). All anti-HIV-1 bNAbs reference antibodies were functionally validated through neutralization assays against the global HIV-1 pseudovirus panel. The resulting $IC_{50}$ and $IC_{80}$ values for each antibody were compared to historical data available in the CATNAP database. Only those antibodies exhibiting less than a threefold deviation in $IC_{50}/IC_{80}$ values relative to the reference data were included in subsequent functional analyses.

007 and 10-1074 IgGs used for structural studies, mass spectrometry, and molar neutralization ratio assays were expressed via transient co-transfection of Expi293F cells with heavy and light chain plasmids. IgG1 antibodies were purified from cell culture supernatant using MabSelect SuRe (Cytiva), concentrated, and SEC purified on a Superdex 200 column (Cytiva). 007 IgG3 was purified by diluting the Expi293F cell culture supernatant 5-fold in PBS (pH 7.0) and loading over a HiTrap Protein G HP column (Cytiva). The column was washed in PBS (pH 7.0) and IgG3 was eluted in 0.1 M glycine, 150 mM NaCl (pH 2.7) into 2 M Tris (pH 8.0). The IgG3 sample was then concentrated and SEC purified on a Superdex 200 column (Cytiva). SEC fractions corresponding to IgG were combined and concentrated.

To produce Fabs, the heavy chain variable region of 007 was subcloned into a mammalian expression vector containing the CH1 domain and a C-terminal 6xHis tag. 10-1074 plasmids were cloned as previously described[25]. Fab heavy and light chain plasmids were used to transiently co-transfect Expi293F cells (Thermo Fisher Scientific) and Fabs were purified from culture supernatant by immobilized metal affinity chromatography (IMAC) using a HisTrap HP column (Cytiva). Fabs were concentrated and buffer exchanged into TBS (20 mM Tris pH 8.0, 150 mM NaCl) using Amicon 10 kDa spin concentrators (Millipore) and further purified by SEC on a Superdex 200 column (Cytiva) equilibrated with TBS. SEC fractions corresponding to Fab were combined and concentrated.

To produce bispecific IgGs comprising a 007 arm and an anti-CD3 OKT3 arm[31], mutations were introduced into the respective IgG1-LS[61] constant regions (E357Q and S364K in $007_{HC}$ and Q295E, L368D, K370S, N384D, Q418E, N421D in $OKT3_{HC}$) to promote heavy chain heterodimerization and facilitate downstream purification[62], and the CH1 and CL domains of the OKT3 antibody were domain swapped to promote correct heavy and light chain pairing of both Fab arms[63]. Heavy and light chain plasmids were mixed in equal amounts (50 μg of each of the four plasmids per 200 ml transfection culture) and used to transiently co-transfect Expi293F cells (Thermo Fisher Scientific). IgGs were purified from cell culture supernatant using a MabSelect SuRe (Cytiva) affinity column and were concentrated and buffer exchanged into 50 mM Tris, pH 8.7 using 10 kDa Amicon spin concentrators (Millipore). Bispecific

IgGs were purified by anion exchange chromatography using a HiTrap Q column (Cytiva) and eluted with a NaCl gradient. Fractions corresponding to the bispecific IgG were combined and concentrated before SEC purification using a Superdex 200 column (Cytiva) equilibrated in TBS.

### Expression and purification of BG505 SOSIP trimers
BG505 (ref. 20) and BG505-DS[64,65] SOSIPs used for cryo-EM, mass spectrometry and SPR experiments were expressed via transient co-transfection of Expi293F cells with a plasmid encoding soluble furin. Briefly, SOSIPs were purified from cell culture supernatant by either PGT145 or 2G12 immunoaffinity chromatography, dialyzed in TBS, concentrated to <2 ml and purified by SEC on a Superose 6 Increase column (Cytiva). SEC fractions corresponding to trimeric SOSIPs were combined and concentrated.

### Quantification of unpurified mAbs from cell supernatants by human IgG capture ELISA
A human IgG capture ELISA was employed to measure antibody concentrations in unpurified supernatants from transfected HEK293-6E cells, with slight modifications to established protocols[21]. ELISA plates (Greiner Bio-One) were coated with 2.5 μg ml$^{-1}$ polyclonal goat anti-human IgG (Jackson ImmunoResearch) in PBS and incubated for at least 45 min at 37 °C or alternatively overnight at 4 °C. Following the coating step, plates were blocked for 60 min at room temperature (RT) with a blocking buffer (BB) composed of PBS supplemented with 5% nonfat dry milk powder (Carl Roth, T145.2). Supernatants from transfected HEK293-6E cells were diluted 1:20 in BB before analysis, while a human myeloma IgG1 kappa standard (Sigma-Aldrich) was prepared at an initial concentration of 4 μg ml$^{-1}$ in BB. Both samples and standards were subjected to serial 1:3 dilutions in BB and incubated for 45 min at RT. Detection was performed using an horseradish peroxidase-conjugated anti-human IgG antibody (Southern Biotech 2040-05) diluted 1:2,500 in BB. Colorimetric development was initiated by the addition of ABTS substrate (Thermo Fisher Scientific, 002024), and absorbance was measured at 415 nm and 695 nm using a microplate reader (Tecan). Antibody concentrations in the supernatants were calculated by interpolation from the human IgG1 standard curve.

### Generation of HIV-1 pseudoviruses
HIV-1 pseudoviruses were produced in HEK293T cells by co-transfection with pSG3Δenv (NIBSC, #2003) and the respective HIV-1 Env plasmids as previously described[19,23,39,66]. A synthetic HIV-1$_{ADA}$ Env plasmid was obtained from Twist Bioscience for the production of HIV-1$_{ADA}$ pseudoviruses containing point mutations identified through in vivo single-genome sequencing (SGS) analysis. In HIV-1 BG505$_{T332N}$ and Tro11 site-specific mutations were introduced using the Q5 Site-Directed Mutagenesis Kit (New England Biolabs) according to the manufacturer's instructions. Kifunensine-treated pseudoviruses were produced under presence of 5 μg ml$^{-1}$ Kifunensine. Supernatants containing Kifunensine-treated pseudoviruses were harvested 3 days after transfection and stored at −80 °C.

### Generation of mutant HIV-1 pseudoviruses
Mutant variants of HIV-1 pseudoviruses were generated by introducing site-specific mutations into gp160 expression plasmids. Point mutations were incorporated using the Q5 Site-Directed Mutagenesis Kit (New England Biolabs) following the manufacturer's protocol. The resulting mutant plasmids were subsequently used for pseudovirus production, following the protocol as described above for wild-type pseudoviruses[66].

### Determination of neutralizing activity by luciferase-based TZM-bl assays
Neutralization assays were performed to determine the $IC_{50}$ and $IC_{80}$ values of purified mAbs and to assess neutralizing activity in unpurified

HEK293-6E cell culture supernatants. These assays were performed with slight modifications to previously described protocols[66–68]. Purified mAbs, serum IgGs, or HEK293-6E cell culture supernatants were preincubated with HIV-1 pseudovirus strains for 1 h at 37 °C before adding $10^4$ TZM-bl cells per well in a 96-well plate. After 48 h of incubation at 37 °C with 5% $CO_2$, luciferase activity was measured using a luciferin/lysis buffer. Background RLUs from non-infected control wells were subtracted, and the percentage of neutralization was calculated. For screening unpurified IgGs in HEK293-6E cell supernatants, a final IgG concentration of 2.5 µg ml$^{-1}$ was used, as determined by human IgG capture ELISA. For large-scale donor screening, IgGs isolated from participant samples were tested against each pseudovirus at a fixed concentration of 300 µg ml$^{-1}$ in duplicate wells. $IC_{50}$ and $IC_{80}$ values for mAbs were determined by performing serial dilutions starting at 10, 25 or 50 µg ml$^{-1}$. These values, representing the mAb concentration required to reduce viral signal by 50% or 80%, were calculated using a dose-response curve fitted in GraphPad Prism. All $IC_{50}$ and $IC_{80}$ determinations were performed in duplicates for each mAb.

## MNR assays

Pseudovirus neutralization assays were conducted using TZM-bl reporter cells as above and as previously described[67,68]. IgGs, bispecifics, and Fabs were evaluated in duplicate with an eight-point, fivefold dilution series starting at a top concentration of 2, 5, or 50 µg ml$^{-1}$. 007 antibodies were expressed and purified within 3 weeks of neutralization assays. The dilution at which 50% of virus was neutralized ($IC_{50}$) is reported in micrograms per milliliter and molar concentrations in Supplementary Table 10. MNRs were calculated as the ratio of molar $IC_{50}$ values for different formats of the antibody.

## Determination of antibody interference by competition ELISAs

To assess antibody interference, mAbs were biotinylated using the EZ-Link Sulfo-NHS-Biotin Kit (Thermo Fisher Scientific) according to the manufacturer's protocol. Excess biotin was removed by buffer exchange into PBS using Amicon 10 kDa centrifugal filter membranes (Millipore). High-binding ELISA plates (Greiner Bio-One) were coated overnight at 4 °C with an anti-6×His tag antibody (Abcam, 9108) at a final concentration of 2 µg ml$^{-1}$. After coating, plates were blocked for 1 h at 37 °C with PBS supplemented with 3% bovine serum albumin (BSA; Sigma-Aldrich) to prevent nonspecific binding. BG505$_{SOSIP.664}$-His protein was then added at 2 µg ml$^{-1}$ in PBS and incubated for 1 h at RT to facilitate antigen capture. To evaluate competition, unlabeled antibodies were applied at an initial concentration of 32 µg ml$^{-1}$ in PBS and subjected to a 1:3 serial dilution. After a 1-h incubation at RT, biotinylated mAbs were introduced at 0.5 µg ml$^{-1}$ in PBS containing 3% BSA, followed by another 1-h incubation at RT. Detection was performed using peroxidase-conjugated streptavidin (Jackson ImmunoResearch) diluted 1:5,000 in PBS containing 1% BSA and 0.05% Tween-20. Between each incubation step, wells were thoroughly washed with PBS containing 0.05% Tween-20 (Carl Roth) to remove unbound components. Colorimetric detection was achieved using ABTS substrate solution (Thermo Fisher Scientific 002024), and absorbance was measured at 415 nm and 695 nm using a microplate reader (Tecan).

## Neutralization fingerprint analyses

The neutralizing activity against the f61 pseudovirus[19] panel was analyzed by calculating Spearman correlation coefficients for each pair of antibodies based on their neutralization data, and visualizing the results as a heatmap (Fig. 1d). Neutralization fingerprint analysis (Fig. 1e) was performed based on a diverse panel of 119 HIV-1 strains for which $IC_{50}$s were available for all antibodies using an approach described previously [69]. An antibody-antibody distance matrix was calculated pairwise as the sum over the panel of the absolute differences of the log $IC_{50}$s. The dendrogram was calculated from this distance

matrix using hierarchical clustering by the R command "hclust" using the "average" method.

## Assessment of autoreactivity in HEp-2 cell assays

Autoreactivity of mAbs was evaluated using the NOVA Lite HEp-2 ANA Kit (Inova Diagnostics) following the manufacturer's protocol. Antibodies were applied at a final concentration of 100 µg ml$^{-1}$ in PBS to HEp-2 cell-coated slides. After incubation and subsequent washing steps, fluorescence imaging was performed using a DMI 6000 B fluorescence microscope (Leica) under standardized conditions: a 3-s exposure time, 100% light intensity and a gain setting of 10. Fluorescent signals were analyzed to determine autoreactivity profiles.

## Cryo-EM sample preparation

007 Fab was incubated in a 3.4:1 molar ratio of Fab to BG505-DS SOSIP trimer[64,65] and incubated overnight at room temperature. SOSIP trimers and Fab-SOSIP complexes were purified from unbound Fabs on a Superose 6 10/300 Increase column (Cytiva) operating in TBS and concentrated using a 30 kDa spin concentrator to ~3.4 mg ml$^{-1}$ immediately before vitrification. 007 IgG1 was added to BG505 SOSIP at a 1:1 molar ratio of IgG to BG505 SOSIP trimer, with a total protein concentration of ~2.6 mg ml$^{-1}$ and vitrified after incubating for ~38 h at room temperature.

Octyl-maltoside, fluorinated solution (Anatrace) was added to 0.02% (w/v) final concentration for each sample immediately before addition of 3 µl to a Quantifoil R1.2/1.3 Cu 300 mesh grid (Electron Microscopy Sciences) that had been glow discharged for 1 min at 20 mA using a PELCO easiGlow (Ted Pella). Grids were blotted for 3 to 4 s with Whatman No. 1 filter paper and vitrified in liquid ethane using a Mark IV Vitrobot (Thermo Fisher Scientific) operating at 22 °C and 100% humidity.

## Cryo-EM data collection and processing

Data for the 007 Fab-SOSIP sample were collected on a 300 keV Titan Krios transmission electron microscope (Thermo Fisher) equipped with a Gatan BioQuantum Energy Filter and a K3 6k x 4k direct electron detector, and data for the 007 IgG1-SOSIP sample were collected on a 200 keV Talos Arctica (Thermo Fisher Scientific) equipped with a Gatan K3 6k x 4k direct electron detector. 40-frame movies were collected in SerialEM[70]. 007 Fab-SOSIP movies were recorded in super-resolution (0.416 Å per pixel) using a 3 × 3 beam image shift pattern with 3 shots per hole and 007 IgG-SOSIP movies were recorded in super-resolution (0.72 Å per pixel) with a 3 × 3 beam image shift pattern and 1 shot per hole.

Data collection and processing details are included in Supplementary Table 8 and Extended Data Figs. 3 and 7. Motion correction, CTF estimation, particle picking and particle extraction were performed in cryoSPARC Live v4 (ref. 71). Extracted particles were subject to 2D classification in cryoSPARC[71] and particles from select 2D classes processed by 3D classification in RELION v4.0.1 (refs. 72,73). Particles from select 3D classes were re-extracted in cryoSPARC and subject to ab initio reconstruction and non-uniform refinement[71,74]. Particles underwent reference-based motion correction and were subject to a final round of non-uniform refinements[74].

## Model building and refinement

Initial coordinates for the 1 Fab-bound trimer structure were generated by docking individual protein chains from reference structures (PDB 5BZD ($V_H$), 7PS3 ($V_L$), and 6UDJ (BG505)) into the corresponding EM density in ChimeraX[75,76]. The initial model was sequence corrected in Coot[77] and underwent iterative rounds of refinement in Phenix[78] and Coot[77]. Glycans were built in Coot[79] and glycan geometries evaluated in Privateer[80]. Coordinates for the 1 Fab-bound trimer aided in generating trimer structures with 0, 2 or 3-bound Fabs, as well as the 3 IgG-bound trimer-dimer structure. To facilitate measurements between the

C-termini of the Fab heavy chains, the Fab $C_H C_L$ domains from PDB 8UKI were docked into the corresponding densities in the 2 Fab-bound trimer structure and the 3 IgG-bound trimer-dimer structure. Antibodies were numbered according to Kabat.

## Structural analyses

Figures were prepared using UCSF ChimeraX v1.9 (refs. 75,76). 007 Fab-Env interactions were evaluated using the 1 Fab-bound trimer structure. Distance measurements between C termini of Fab HCs were taken from the alpha carbon of R222 in either the 2 Fab-bound trimer structure or the 3 IgG-bound trimer-dimer structure. Coordinates for a 007-bound gp120 were aligned to two gp120 chains in a b12-bound trimer structure (PDB 5VN8)[81] to model the analogous distance on an occluded-open conformation of the trimer. The distance between trimer apexes reported for the trimer-dimer structure was measured between residues N188 on opposing trimers.

## SPR

SPR measurements were performed on a Biacore T200 (GE Healthcare) at 25 °C in HBSEP+ (10 mM Hepes, 150 mM NaCl, 3 mM EDTA and 0.005% Tween-20) (GE Healthcare) running buffer. BG505 and BG505-DS SOSIPs were directly immobilized on a CM5 chip (GE Healthcare) to ~500 resonance units using primary amine chemistry. Fab samples were injected over the flow cells at increasing concentrations (threefold dilution series with a top concentration of 10 μM or 1 μM) at a flow rate of 60 μl min$^{-1}$ for 60 s and allowed to dissociate for 300 s. Regeneration of flow cells was achieved by injecting one pulse of 10 mM glycine pH 2.0 at a flow rate of 90 μl min$^{-1}$. All samples were performed in duplicate and a representative sensorgram is plotted in Extended Data Fig. 6a. Kinetic constants and affinities were derived using the Biacore T200 Evaluation Software (v2.0) with a 1:1 binding model.

## Immunoprecipitation for site-specific N-glycan analyses

To isolate BG505 trimers recognized by 007$_{IgG}$ (007$_{IgG}$-bound BG505), we developed an optimized protocol for immunoprecipitation. Briefly, BG505 and 007$_{IgG}$ were mixed at a 1:2 (w:w) ratio and incubated overnight at 4 °C in 20 mM sodium phosphate buffer (pH 7.2). 007$_{IgG}$-bound BG505 complexes were isolated using affinity chromatography (Protein G Sepharose; GE Healthcare Life Sciences). Protein G Sepharose was equilibrated with 20 mM sodium phosphate buffer (pH 7.2) and incubated overnight at 4 °C with the preincubated mixture of BG505 and 007$_{IgG}$. 007$_{IgG}$-BG505 complexes captured by Protein G Sepharose were separated from unbound BG505 by centrifugation at 1,000 x g for 3 min, eluted with 0.1 M glycine (pH 2.5) and neutralized with 1 M Tris-HCl (pH 9.0). Samples were then stored at −20 °C.

## Enzymatic removal of N-glycans from Env BG505 SOSIP

Total BG505 and 007$_{IgG}$-bound BG505 samples were denatured for 10 min at 100 °C in a denaturing buffer provided with peptide N-glycosidase F (PNGase F, Prozyme). After samples were chilled on ice for 5 min, PNGase F was added together with detergent following the manufacturer's instructions (PNGase F, Prozyme). Samples were then incubated at 37 °C for 30 h.

## Isolation of gp120 component chains of BG505 SOSIP for LC-MS

Total BG505 and 007$_{IgG}$-bound BG505 samples were separated by SDS-PAGE under reducing conditions on 10% Mini-PROTEAN TGX precast gels (Bio-Rad Laboratory). Gels were briefly washed with water and stained with Bio-safe colloidal Coomassie G-250 Stain (Bio-Rad Laboratories). After destaining, protein bands corresponding to natively glycosylated gp120 (~130 kDa) or samples deglycosylated with PNGase F (~65 kDa) were excised from the gel and stored at −20 °C for LC-MS analyses (Extended Data Fig. 5).

## LC-MS and MS/MS analysis of gp120

The excised bands were digested with trypsin (Promega) and extracted from the gel matrix by use of standard in-gel protease digestion methods[82,83]. The resulting peptide/glycopeptide mixtures were analytically separated on a self-prepared C18 reversed-phase pulled-tip column using a nano-liquid chromatography (nano-LC) system as previously described[82,83]. The eluted glycopeptides were electrosprayed at 2 kV into a dual linear quadrupole ion trap Orbitrap Velos Pro mass spectrometer (Thermo Fisher Scientific). The mass spectrometer was set to switch between a full scan ($400 < m/z < 2,000$) followed by successive MS/MS ($200 < m/z < 2,000$) scans of the 10 most abundant precursor ions using the collision-induced dissociation method.

## Glycopeptide identification and quantitation

Site-specific N-glycan heterogeneity profiles of gp120 from total BG505 and 007$_{IgG}$-bound BG505 samples were determined using a workflow similar to that used before[82,83], which consisted of three main steps.

**Step 1. Initial glycopeptide identification.** LC-MS/MS data for the deglycosylated gp120 were analyzed using the Single Protein Screening and Quantitation workflow in the Pinnacle software (version 1.0.103 Optys Tech Corporation). Identification of peptides containing specific N-glycosylation sites (NGSs) was achieved with a peptide tolerance of 10 ppm in MS1 and an MS/MS tolerance of 0.7 Da. All peptide assignments were validated by visual inspection of the associated MS1 and MS/MS spectra. An increase of 1 Da in the peptide mass value of the deglycosylated gp120 indicated the presence of N-glycosylation site with an attached glycan in the intact gp120 because PNGase F treatment converted each glycosylated Asn into Asp. Site occupancy was calculated based on the areas under the curve (AUC) for the peptide containing the unmodified Asn and the peptide containing the Asp.

**Step 2. Glycopeptide quantitation.** To identify the full range of glycan heterogeneity at specific sites, LC-MS/MS data for the intact gp120 were analyzed using the Targeted Quantitation – Label Free DDA workflow in the Pinnacle software. The search was conducted with a 10-ppm mass accuracy of the three most abundant isotopes with a series of custom peptide and glycopeptide workbooks generated for each NGS. To speed up the validation process and reduce the number of false-positive results, the confirmed deglycosylated peptides from Step 1 were used to limit the retention time window search for each glycopeptide. Once the glycopeptides were identified, the AUC from each glycopeptide was expressed as a percent relative abundance of the total sum of all glycoforms for an NGS.

**Step 3. Visualization of N-glycan heterogeneity at specific sites.** The entire range of glycans at a single site was presented as a stacked bar divided into the relative distributions of the broad N-glycan categories of high-mannose, hybrid, and complex, using the same color scheme as previously described[82]. To better visualize changes in glycoforms between total BG505 and 007$_{IgG}$-bound BG505, a side-by-side bar graph was also included which differentiated each high-mannose glycan but kept the hybrid and complex as single groups. Error bars represent the standard deviations of replicate measurements (three for total BG505 and two for 007$_{IgG}$-bound BG505). Differences between groups were evaluated for statistical significance based on $P$ values calculated using Welch's $t$-test.

## Deep mutational scanning

A previously established lentiviral deep mutational scanning platform was utilized to assess the effects of all mutations on HIV Env escape from antibodies[41,42]. This system employs barcoded pseudoviruses to facilitate comprehensive mutational analysis. The BF520 Env and TRO.11 Env mutant libraries were generated in prior studies[17,52] and were reused in the present investigation. Detailed descriptions of the construction

and composition of these mutant libraries are available in previous reports[41,42]. BF520 and TRO.11 Env function of entry into cells and escape from antibody 10-1074 are published in these prior studies[41,42]. In this study, the BF520 and TRO.11 Env mutant libraries were employed to map escape from antibodies 007, BG18, PGT121 and PGT128. For each mutant library, a small proportion of VSV-G pseudotyped viruses carrying unique barcodes was introduced as an internal infection control, as these viruses are not susceptible to HIV-1-specific antibodies. Each library was incubated for 1 h with antibodies at concentrations ranging from $IC_{90}$ to $IC_{99.9}$, alongside a control incubation without antibody. TZM-bl cells were then infected with the antibody-treated pseudovirus libraries, and 12 h after infection, unintegrated lentiviral genomes were extracted using a miniprep approach. Variant barcodes were subsequently amplified and sequenced following previously established protocols[42].

## Deep mutational scanning data analysis

Deep mutational scanning data were analyzed using dms-vep-pipeline-3 version 3.20.1. See https://github.com/dms-vep/HIV_Envelope_TRO11_DMS_007 and https://github.com/dms-vep/HIV_Envelope_BF520_DMS_007 for GitHub repositories containing the analyses of the deep mutational scanning data. See https://dms-vep.org/HIV_Envelope_TRO11_DMS_007 and https://dms-vep.org/HIV_Envelope_BF520_DMS_007 for HTML renderings of key analyses, plots and data files produced by the analyses. Effects of mutations on Env function of entry into cells measured in prior studies[41,42] were used to shade the logoplots in Fig. 5 and Extended Data Fig. 8c Illumina sequencing data of variant barcodes from antibody selection experiments were processed following previously established methods[42]. A comprehensive description of sequencing analyses and modeling of mutation effects is provided in Radford et al.[42], which should be consulted for methodological details. Briefly, the non-neutralized fraction of each barcoded variant was calculated in each antibody selection by comparing the frequency of each barcoded variant to the frequency of the non-neutralized standard viruses between the antibody incubation and mock incubation conditions. The software package polyclonal[84] version 6.14 was used to model the escape effects of individual mutations for each antibody. In Fig. 5a and Extended Data Fig. 8c the height of each amino acid in each logo plot represents the effect of that individual mutation on escape from that antibody, where taller letters represent more escape. The model of the effects of mutations on escape can also be used to predict the level of neutralization of Env mutants at arbitrary antibody concentrations, which we used to calculate the deep mutational scanning measured log fold change in $IC_{50}$ values in Fig. 5b.

## FcγRIIIa signaling Promega assay

Signaling through FcγR was assessed using an ADCC reporter bioassay from Promega (Promega, G9790 (FcγRIIIa-F), G7010(FcγRIIIa-V)). CHO cells overexpressing HIV JRFL envelope (CHO-HIV JRFL ENV) were used as targets. HIV bnAbs were serially diluted 4-fold and added to target cells, 30 min before addition of Jurkat reporter cells. After overnight incubation at 37 °C, luminescence was measured using the Bio-Glo-TM Luciferase Assay Reagent according to the manufacturer's instructions.

## ADCC assay

NK cells were isolated from whole blood using the EasySep Direct Human NK Cell Isolation Kit (StemCell Technologies, #19665) following the manufacturer's instructions. CHO-HIV JRFL ENV were incubated with titrated HIV bnAbs prior to addition of effector NK cells at 8:1 (FV NK donor) or 9:1 (FF NK donor) in AIM V Medium (Life Technologies, #12055091). Each Ab concentration was tested in duplicate. After 4-h incubation at 37 °C, plate was centrifuged and supernatants transferred to a new plate containing the LDH substrate prepared according to manufacturer's instructions (Roche, #11644793001 Cytotoxicity Detection Kit (LDH)). Spontaneous release was determined by incubating

target cells with medium alone, and maximal lysis by incubation with 1% Triton X-100. Kinetic Absorbance was measured at 492 and 650 nm with a microplate reader, and cytotoxicity was determined using the following equation:

$$\text{Percent killing}$$
$$= (\text{experimental} - \text{spontaneous})/(\text{maximal} - \text{spontaneous}) \times 100$$

## Generation of replication-competent HIV-1 virus

Recombinant, replication-competent HIV-1$_{ADA}$ variant, incorporating the ADA Env within an NL4-3 backbone, was produced by transfecting HEK293T cells using FuGENE 6 Transfection Reagent (Promega). Virus-containing supernatants were harvested at 48 and 72 h after transfection, then aliquoted and stored at −80 °C for subsequent experiments.

## HIV-1 infection of humanized mice and viral load quantification

Humanized NRG mice were infected intraperitoneally with replication-competent HIV-1$_{ADA}$. At 32 days post-infection, sterile monoclonal antibodies were administered subcutaneously in PBS. A loading dose of 1 mg was initially delivered, followed by maintenance doses of 0.5 mg every 3 to 4 days. To determine viral loads, viral RNA was extracted from EDTA-treated plasma samples using the MinElute Virus Spin Kit (Qiagen), with DNase I treatment (Qiagen) performed on an automated Qiacube system (Qiagen). pol-specific primers 5′-TAATGGCAGCAATTTCACCA-3′ and 5′-GAATGCCAAATTCCTGCTTGA-3′, along with the probe 5′/56-FAM/CCCACCAACARGCRGCCTTAACTG/ZenDQ/, as previously described[85]. qRT-PCR was performed using a QuantStudio 5 instrument (Thermo Fisher Scientific) with the TaqMan RNA-to-CT 1-Step Kit (Thermo Fisher Scientific). Heat-inactivated supernatants of replication-competent HIV-1$_{YU2}$ propagated in MOLT-CCR5 cells, were included in each PCR run as a standard for viral load quantification. The viral concentration of this standard was determined using the Cobas 6800 HIV-1 kit (Roche). The quantification limit for qRT-PCR was established at 260 copies ml$^{-1}$. To calculate log$_{10}$ changes in viral load, values below this threshold were assigned a concentration of 260 copies ml$^{-1}$. For normalization, the mean log$_{10}$ viral load change of the PBS control group at each corresponding time point was subtracted from the individual log$_{10}$ viral load changes of the treatment groups.

## Single-genome sequencing of plasma HIV-1 env from in vivo experiments

Single-genome sequencing (SGS) of HIV-1 env genes was conducted following previously established protocols[86]. Briefly, viral RNA was extracted from EDTA-treated plasma samples using the MinElute Virus Spin Kit (Qiagen), followed by DNase I treatment (Qiagen) on an automated Qiacube system (Qiagen). Reverse transcription was performed using the antisense primer YB383 (5′-TTTTTTTTTTTTTTTTTTTTTTTTTTRAAGCAC-3′) and SuperScript IV reverse transcriptase (Thermo Fisher Scientific) according to the manufacturer's instructions. To degrade residual RNA, the reaction was treated with 0.25 U µl$^{-1}$ RNase H (Thermo Fisher Scientific) at 37 °C for 20 min. The resulting cDNA encoding HIV-1 Env was subjected to serial dilution before nested PCR amplification using PlatinumTaq Green HotStart polymerase (Thermo Fisher Scientific) with HIV-1$_{ADA-NL4-3}$-specific primers. The first-round PCR utilized primers YB383 and YB50 (5′-GGCTTAGGCATCTCCTATGGCAGGAAGAA-3′) with the following thermocycling conditions: an initial denaturation at 94 °C for 2 min, followed by 35 cycles of 94 °C for 30 s, 55 °C for 30 s and 72 °C for 4 min, with a final extension at 72 °C for 15 min. The second-round PCR was performed using 1 µl of the first-round product as the template, with primers YB49 (5′-TAGAAAGAGCAGAAGACAGTGGCAATGA-3′) and YB52 (5′-GGTGTGTAGTTCTGCCAATCAGGGAAGWAGCCTTGTG-3′).

The cycling conditions were similar to the first PCR, except the reaction was run for 45 cycles. PCR products from reactions demonstrating an amplification efficiency of less than 30% were selected for Sanger sequencing.

### In vivo pharmacokinetic analysis

To determine the half-life of administered antibodies, human FcRn transgenic mice (B6.Cg-Fcgrt^tm1Dcr Prkdc^scid Tg(FCGRT)32Dcr/DcrJ, Jackson Laboratory; $n = 12$, all female) (The Jackson Laboratory) were injected intravenously with 0.5 mg sterile antibody in PBS via the tail vein. Human IgG concentrations in serum were quantified using an ELISA assay with slight modifications to previously established methods[51]. For quantification, high-binding ELISA plates (Corning) were coated overnight at RT with 2.5 µg ml⁻¹ of anti-human IgG (Jackson ImmunoResearch). Plates were then blocked for 1 h at RT with a blocking buffer containing 2% BSA (Carl Roth), 1 µM EDTA (Thermo Fisher Scientific), and 0.1% Tween-20 (Carl Roth) in PBS to minimize nonspecific binding. A standard curve was generated using human IgG1 kappa purified from myeloma plasma (Sigma-Aldrich) and serially diluted in PBS. Serum or plasma samples, along with standard dilutions, were incubated on the plates for 90 min at RT. This was followed by a 90-minute incubation with horseradish peroxidase-conjugated anti-human IgG (Jackson ImmunoResearch) at a 1:1,000 dilution in blocking buffer. Colorimetric detection was performed using ABTS substrate (Thermo Fisher Scientific), and absorbance was measured at 415 nm using a Tecan microplate reader. Each step included washing with 0.05% Tween-20 in PBS. Baseline serum or plasma samples confirmed the absence of human serum or plasma IgG before antibody injection.

### Quantification and statistical analysis

Flow cytometry data were processed and analyzed using FlowJo v10 software. Statistical analyses were conducted using GraphPad Prism (versions 7 and 8), Python (v3.6.8), R (v4.0.0) and Microsoft Excel for Mac (versions 14.7.3 and 16.4.8).

For clonal sequence evaluation, complementarity-determining region 3 of the heavy chain (CDRH3) lengths, V gene usage and germline identity distributions were assessed across all input sequences without additional sequence collapsing.

### Use of large language models

ChatGPT (v.4 and v.5) was used for general editorial tasks, including proofreading, grammar correction and text summarization. Scientific content and conclusions are the work of the authors.

### Inclusion and ethics

Research has been conducted and authorships have been determined in alignment with the Global Code of Conduct for Research in Resource-Poor Settings.

### Reporting summary

Further information on research design is available in the Nature Portfolio Reporting Summary linked to this article.

## Data availability

Nucleotide sequences of bnAb 007 and clonal members have been deposited at GenBank under accession codes PX523309 - PX523316 and at European Nucleotide Archive under accession codes ERR15846755 - ERR15846755 and ERR15881563 - ERR15881564 under the study PRJEB102783. The NGS B cell repertoire data analyzed in this study have been deposited in the Sequence Read Archive (SRA) under accession codes SAMN29624595 to SAMN29624713 [https://www.ncbi.nlm.nih.gov/sra?linkname=bioproject_sra_all&from_uid=857338] and the BioProject database under accession code PRJNA857338. Cryo-EM maps and models have been deposited in the Electron Microscopy Data Bank (EMDB) and Protein Data Bank with accession codes: EMD-70018 and PDB ID 9O2Q (007-BG505v2, 0 Fabs bound), EMD-70019 and PDB ID 9O2R (007-BG505v2, 1 Fab bound), EMD-70020 and PDB ID 9O2S (007-BG505v2, 2 Fabs bound), EMD-70021 and PDB ID 9O2T (007-BG505v2, 3 Fabs bound), EMD-70022 and PDB ID 9O2U (007-BG505v2, IgG crosslinked trimer-dimer). Neutralization data of 007 have been deposited at CATNAP database. Clinical data are available upon request to the corresponding author (F.K.) provided that there is no reasonable risk of deanonymizing study participants and/or may require a Material Transfer Agreement (MTA). Individual donor data cannot be shared due to privacy restrictions. Requests will be responded to within 2 weeks. Source data are provided with this paper.

## Code availability

All computer code and data for the deep mutational scanning is publicly available on GitHub (https://dms-vep.org/HIV_Envelope_TRO11_DMS_007/htmls/all_antibodies_and_cell_entry_overlaid.html).

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

## Acknowledgements

We thank all study participants for supporting our research by blood donation; members of the Klein, Bjorkman and Bloom Labs and S. Rose for their support and inspiring discussions; and A. Schmitt and T. Bresser for lab management and assistance. We thank the staff of the Animal Care Facility Weyertal at the University of Cologne. We thank S. Chen and the Beckman Institute Resourse Center for Transmission Electron Microscopy at Caltech, as well as J. Vielmetter and the Caltech Protein Expression Center. We thank P. Gnanapragasam for help conducting MNR neutralization assays. A.T.D. was supported by an NSF Graduate Research Fellowship. The panel of global HIV-1 pseudoviruses was obtained through the NIH AIDS Reagent Program. This work was funded by grants from the German Center of Infection Research (DZIF) to F.K., the German Research Foundation (DFG) CRC1310 and European Research Council (ERC) ERC-stG639961 to F. K. This work was also supported, in whole or in part, by the Bill & Melinda Gates Foundation (grant INV-002143 to F.K. and P.J.B. and grant INV-036842 to M.S.S.). Under the grant conditions of the Foundation, a Creative Commons Attribution 4.0 Generic License has already been assigned to the Author Accepted Manuscript version that might arise from this submission. Research reported in this publication was also supported by the National Institute of Allergy and Infectious Diseases of the National Institutes of Health under award numbers P01AI100148 and 1U54AI170856 to P.J.B. and R01AI140891 and U01AI169385 to J.D.B. Z.Y., Q.W., M.B.R., T.J.G. and J.N. are supported in part by a grant from National Institutes of Health (AI162236) and research-acceleration funds from the University of Alabama at Birmingham. The content is solely the responsibility of the authors and does not necessarily represent the official views of the National Institutes of Health. P.S. is supported by the Emmy Noether Programme of the German Research Foundation (DFG; project 495793173). BNAb 007 has been exclusively licensed to Vir Biotechnology.

## Author contributions

Conceptualization was carried out by L.G., A.T.D., M.R., H.B.G., P.J.B. and F.K. Methodology was developed by L.G., C.K., H.G., A.T.D., H.B.G., C.R., J.D.B., Z.Y., Q.W., M.B.R., T.J.G., J.N., P.S., M.S.S., P.J.B. and F.K. Formal analysis was performed by L.G., A.T.D., M.R., C.K., H.B.G., E.A., A.P.W., I.S.G., M.J.G., C.R., M.S., J.W., A.A., P.J.B. and F.K. Investigation was conducted by L.G., A.T.D., M.R., M.S.E., H.B.G., C.H.-D., S.D., J.A.B., F.G., D.C., J.K., R.S., C.R., M.S.S., E.A., Z.Y., Q.W., M.B.R., I.S.G., M.J.G., T.J.G., J.N., J.D.B., H.G., P.J.B. and F.K. Resources were provided by A.K., C.G., L.M., W.W., N.E.N. and M.H. L.G., A.T.D., M.R., H.B.G., C.R., C.H.D., J.D.B., P.J.B., H.G. and F.K. wrote the original draft, and all authors wrote, reviewed and edited the paper. Supervision was provided by H.B.G., H.G., P.J.B. and F.K. Funding was aquired by M.S.S., J.D.B., F.K. and P.J.B. L.G., A.T.D. and M.R. contributed equally to this work.

## Funding

## Competing interests

A patent application (Germany Patent application no. EP23203598) incorporating elements of this study has been submitted by the University of Cologne, with L.G. and F.K. listed as inventors. Additionally, H.G., P.S. and F.K. are named inventors on other patent applications related to HIV-1 neutralizing antibodies. L.G., H.G., P.S. and F.K. have received financial compensation from the University of Cologne for licensed patents. C.H.D, S.D, F.G. and D.C. are employees and stockholders of Vir Biotechnology. J.D.B. and C.R. are inventors on patents related to viral deep mutational scanning, licensed by Fred Hutch. J.D.B. also serves as a consultant for Apriori Bio. M.B.R. and J.N. are co-founders and co-owners of Reliant Glycosciences. The other authors declare no competing interests.

## Additional information

**Extended data** is available for this paper at

**Correspondence and requests for materials** should be addressed to Florian Klein.

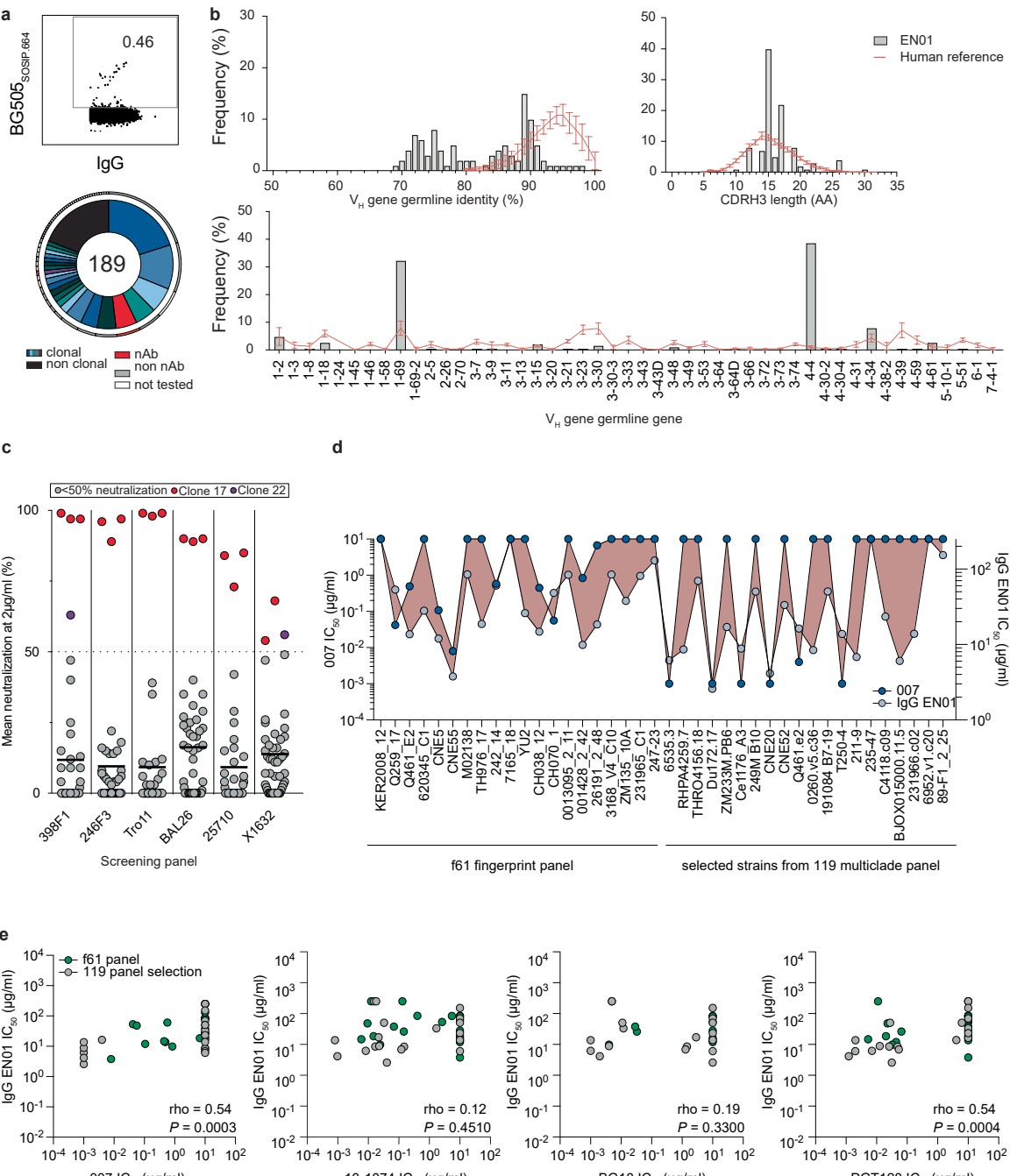

**Extended Data Fig. 1 | Isolation of HIV-1-reactive B cells and analysis of neutralizing serum and antibody response. a**, FACS plots showing the sorting gate and frequencies (in %) of HIV-1-reactive, IgG+ B cells isolated from donor EN01 as previously described[49]. A pie chart illustrates the clonal relationships of amplified heavy chain sequences from single B cells. Individual clones are represented by shades of blue, gray, and white, with the total number of productive IgG heavy chain sequences indicated at the pie chart's center. Clone sizes are proportional to the total number of productive sequences per clone. **b**, VH gene distribution, VH gene germline identity at nucleotide level, and CDRH3 length (in amino acids) for amplified IgG heavy chains compared to a memory IgG reference. The red line indicates the mean ± SD for the reference.

**c**, Neutralizing activity of 48 isolated mAbs against a screening pseudovirus panel consisting of 6 viral strains from 4 different clades. Each mAb was tested at 2 µg/mL, with those achieving >50% neutralization classified as neutralizing. Neutralizing antibodies from B cell clone 17 (red) and clone 22 (purple) are highlighted. **d**, Neutralizing activity of 007 (left y-axis) compared to IgG of donor EN01 (right y-axis) against the f61 fingerprint panel and selected pseudovirus strains from the 119 multiclade panel. Pseudovirus strains are indicated on the x-axis. **e**, Correlation between 007, V3 reference bNAbs and donor serum neutralizing activity against both pseudovirus panels (two-tailed Spearman's rho test).

**a**

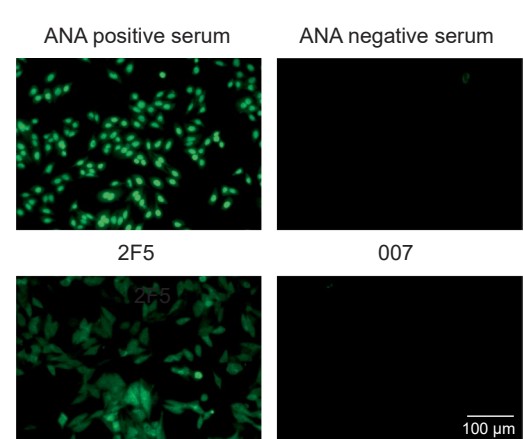

**b**

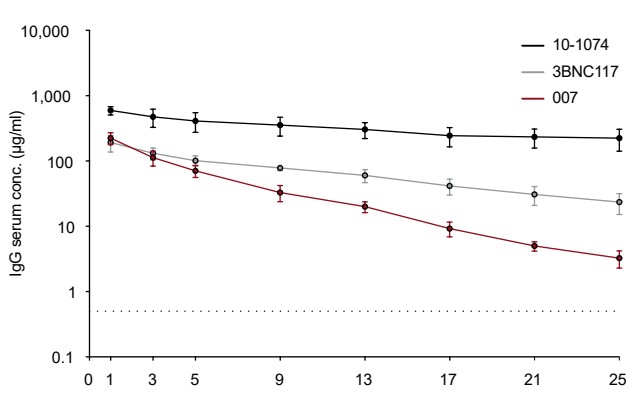

**Extended Data Fig. 2 | Preclinical characteristics of 007. a**, Reactivity of indicated antibodies against HEp-2 cells. Antibodies were tested at a concentration of 100 μg/mL. **b**, In vivo pharmacokinetic profile of 007 and reference bNAbs measured in hFcRn transgenic mice (B6.Cg-Fcgrt^tm1Dcr^Prkdc^sci^ ^d^Tg(FCGRT)32Dcr/DcrJ, Jackson Laboratory; n = 12, all female). Mice received a single intravenous injection of 0.5 mg of antibodies (n = 4 per group). Each dot represents the mean ± S.D. serum concentration for all mice in the group at a given time point, with individual serum samples measured in technical duplicate by enzyme-linked immunosorbent assay (ELISA). Control antibody data were previously reported[49] and were generated alongside the data for 007 in the same experiment (**a** and **b**).

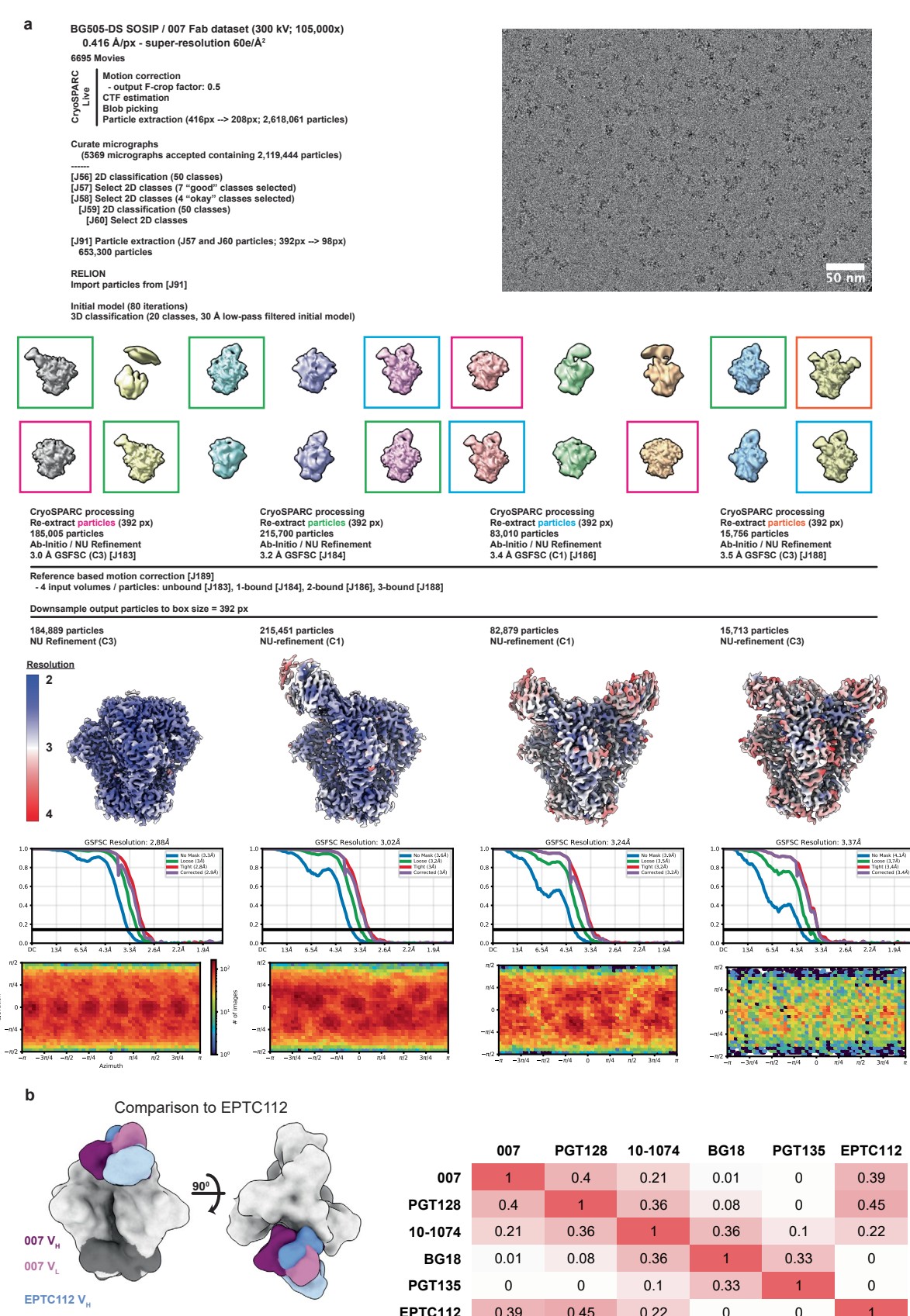

**Extended Data Fig. 3 | See next page for caption.**

**Extended Data Fig. 3 | Data collection and processing for 007 Fab-BG505-DS complexes and comparison of binding poses with V3 reference bNAbs.**
**a**, Example micrograph, data processing workflow, final densities colored by local resolution, gold-standard Fourier shell correlations (GSFSC), and particle orientation distribution plots. **b**, Overlay of 007 with V3-targeting bNAb EPTC112 (PDB code 8C8T) (left). To calculate $V_H$-$V_L$ volume overlap, coordinates (PDB codes 5C7K, 5T3Z, 6CH7, 4JM2) were aligned on gp120 and depicted as solvent-excluded surfaces in ChimeraX[75,76]. The overlap volume between two antibodies was calculated as the sum of the two $V_H$-$V_L$ volumes minus the volume enclosed by the merged $V_H$-$V_L$ surfaces. The Dice similarity coefficient is reported (right).

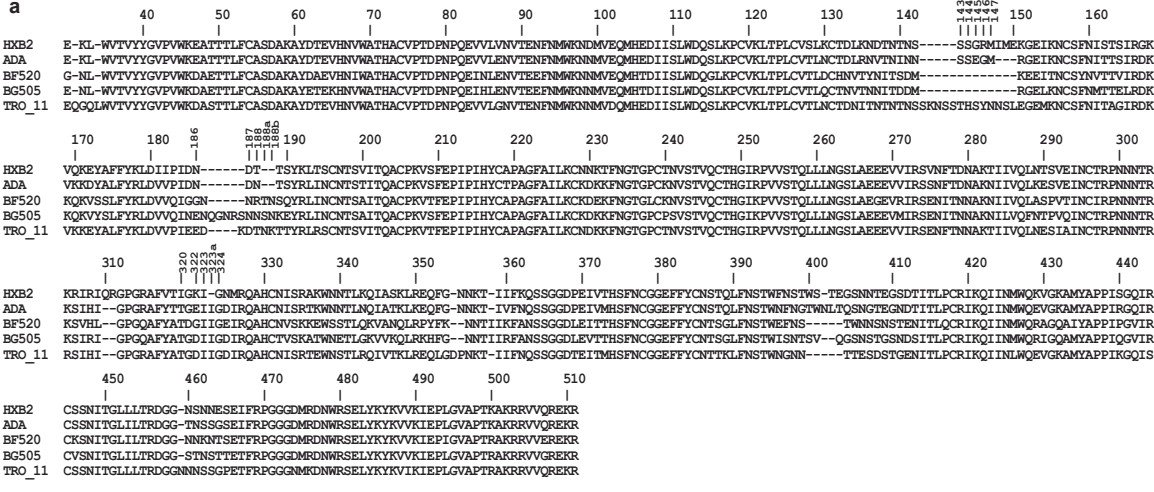

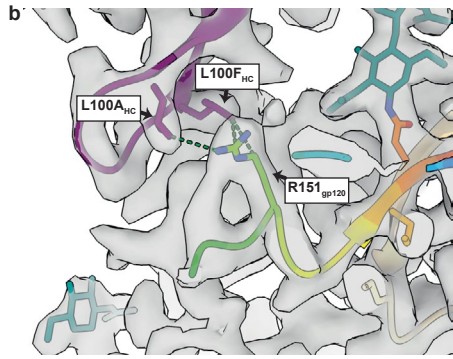

**b**

**c**

gp120 bound by 007 Fab

unbound gp120

**d**

Extended Data Fig. 4 | See next page for caption.

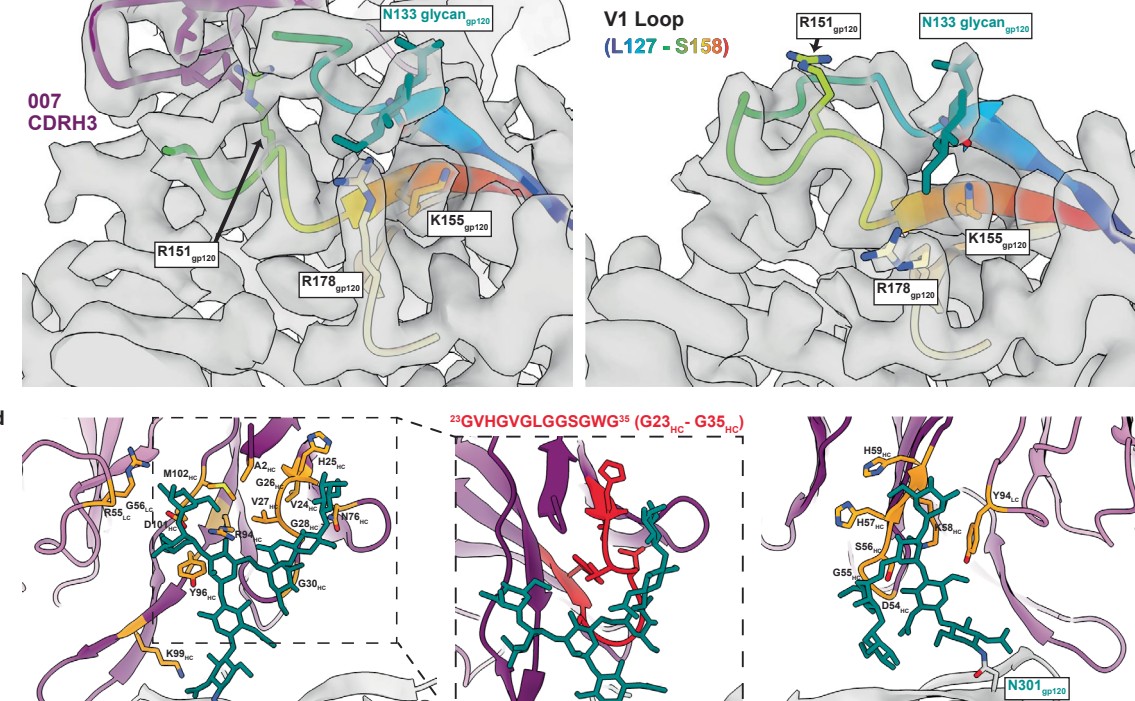

**Extended Data Fig. 4 | Interactions of 007 with the V1 loop and N-glycans on Env. a**, Alignment of HIV-1 *env* sequences. Residues are numbered according to HIV-1$_{HXB2}$. **b**, Contacts between the V1 residue R151$_{gp120}$ and 007. Contacts on 007 within 4 Å of R151$_{gp120}$ are shown by green dotted lines. **c**, Differences in the V1 loop on gp120 between a protomer bound by 007 Fab (left) and an unbound protomer (right). **d**, Contacts between 007 and the N156$_{gp120}$ glycan (left) or the N301$_{gp120}$ glycan (right). 007 residues within 4 Å of modeled glycans are colored orange. Inset (middle) highlights the interactions between a glycine-rich motif in 007, colored red, and the N156$_{gp120}$ glycan.

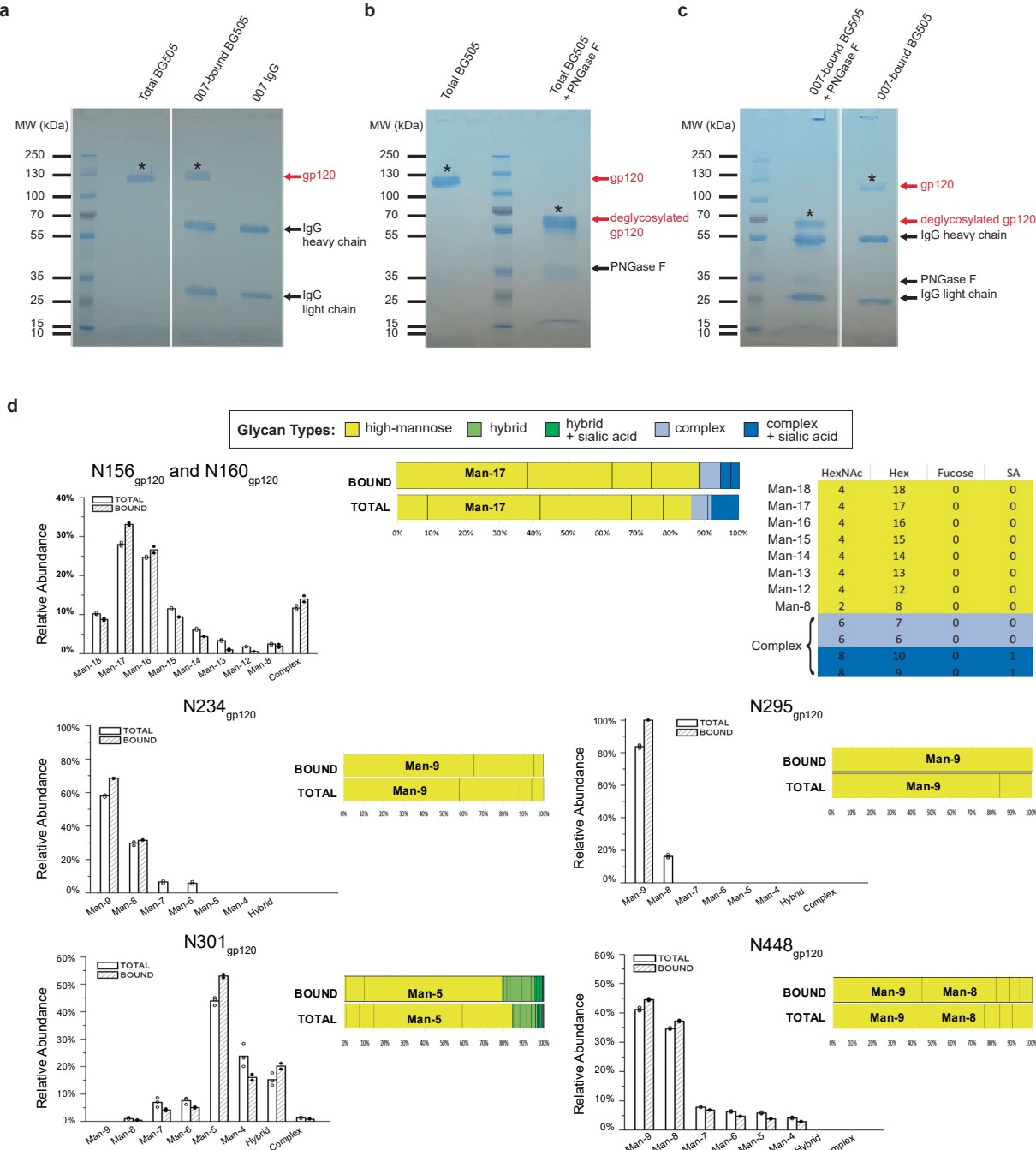

**Extended Data Fig. 5 | N-glycan analysis by quantitative mass spectrometry.**
Electrophoretic separation of gp120 Env subunit using SDS-PAGE under
denaturing and reducing conditions, stained with Coomassie G-250 (**a-c**):
**a**, Total BG505, 007$_{IgG}$-bound BG505, and 007$_{IgG}$ control; **b**, Total BG505 with and
without PNGase F treatment; **c**, 007$_{IgG}$-bound BG505 with and without PNGase
F treatment. The shift in mobility of PNGase F-treated samples in (**b-c**) indicates
removal of N-glycans from gp120. Arrows mark gp120, deglycosylated gp120,
IgG heavy and light chains, and PNGase F. Asterisks indicate bands that were
excised and stored for LC-MS analyses. Red text and arrows indicate gp120 or
deglycosylated gp120 bands. **d**, Comparison of glycoform abundance in total
unliganded BG505 and 007$_{IgG}$-bound BG505 at N156$_{gp120}$/N160$_{gp120}$, N234$_{gp120}$,
N295$_{gp120}$, N301$_{gp120}$, and N448$_{gp120}$. Data are presented as a side-by-side bar
graph, where different high-mannose glycoforms are differentiated, but hybrid
and complex-type glycans are presented as single groups. Points represent the
replicative measurements (three for total BG505 and two for bound BG505) and
bar graphs represent the mean of the replicative measurements. For visualization
purposes, data are also presented as a stacked bar with individual glycoforms
separated by a vertical line, and the most prevalent glycoform(s) labeled.

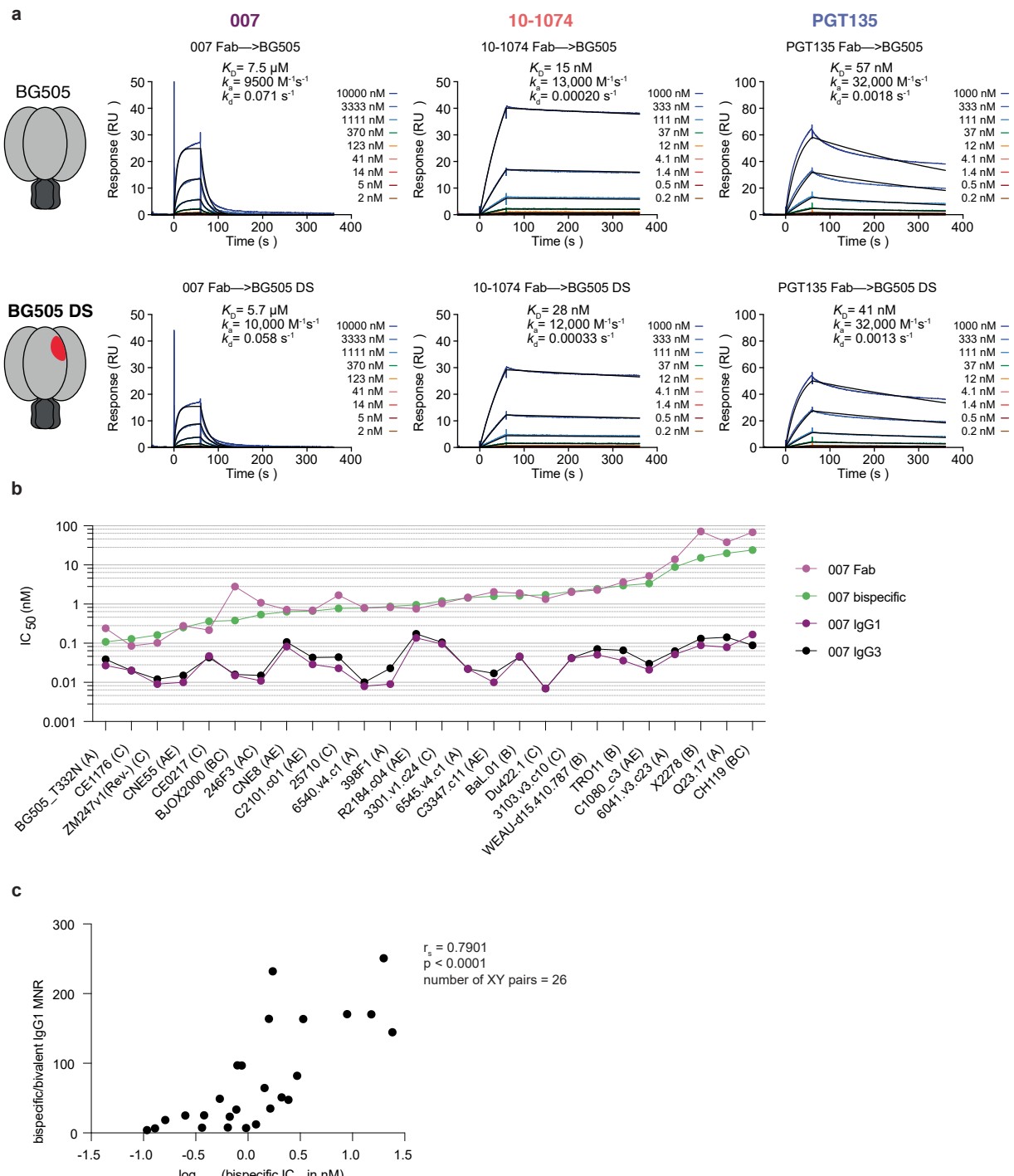

**Extended Data Fig. 6 | Binding affinities and neutralization activities for 007.**
**a**, V3 bNAb Fabs were injected over BG505 (top) or BG505-DS (bottom) SOSIP trimers, and a 1:1 binding model was used to fit the experimental data and derive kinetic constants. Experimental data are shown by colored lines and predicted values are shown by black lines. A representative SPR sensorgram of duplicate experiments is shown. **b**, Molar IC$_{50}$ values for IgG1, IgG3, bispecific IgG1, and Fab forms of 007 against a panel of HIV-1 pseudoviruses. The data are the same as in

Fig. 3b, but rearranged in order of decreasing potency (that is, increasing IC$_{50}$ values) of the 007 bispecific IgG1 to highlight the positive correlation between decreasing monovalent potencies and increasing bispecific / bivalent IgG1 MNRs. **c**, A plot showing the positive correlation between the 007 bispecific IC$_{50}$ values and the bispecific / bivalent IgG1 MNRs. The Spearman correlation coefficient (r$_s$) and p-value (< 0.0001) were determined by a two-tailed Spearman's rho test and are illustrated next to the plot.

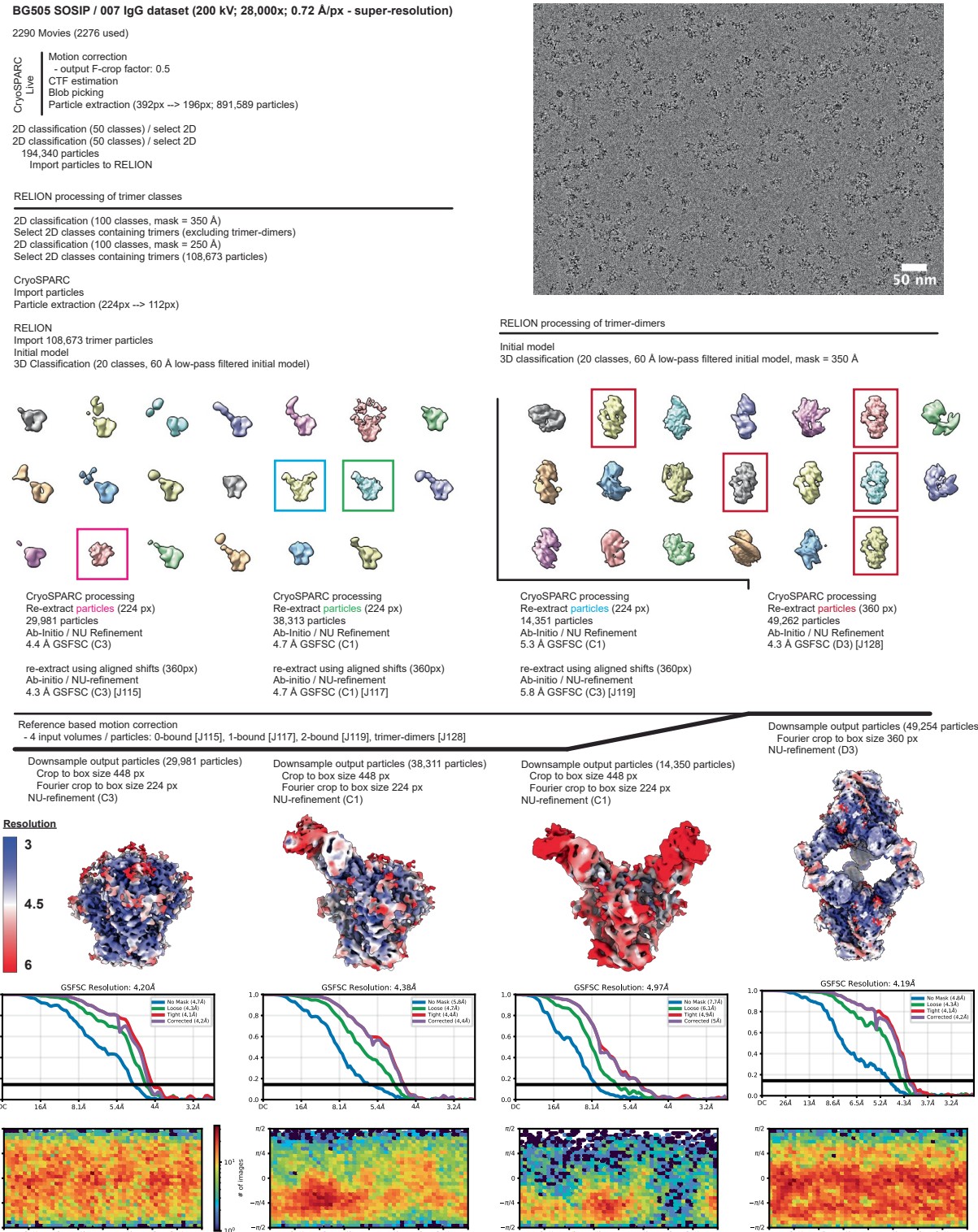

**Extended Data Fig. 7 | Data collection and processing for 007 IgG-BG505 complexes.** Example micrograph, data processing workflow, final densities colored by local resolution, gold-standard Fourier shell correlations (GSFSC), and particle orientation distribution plots.

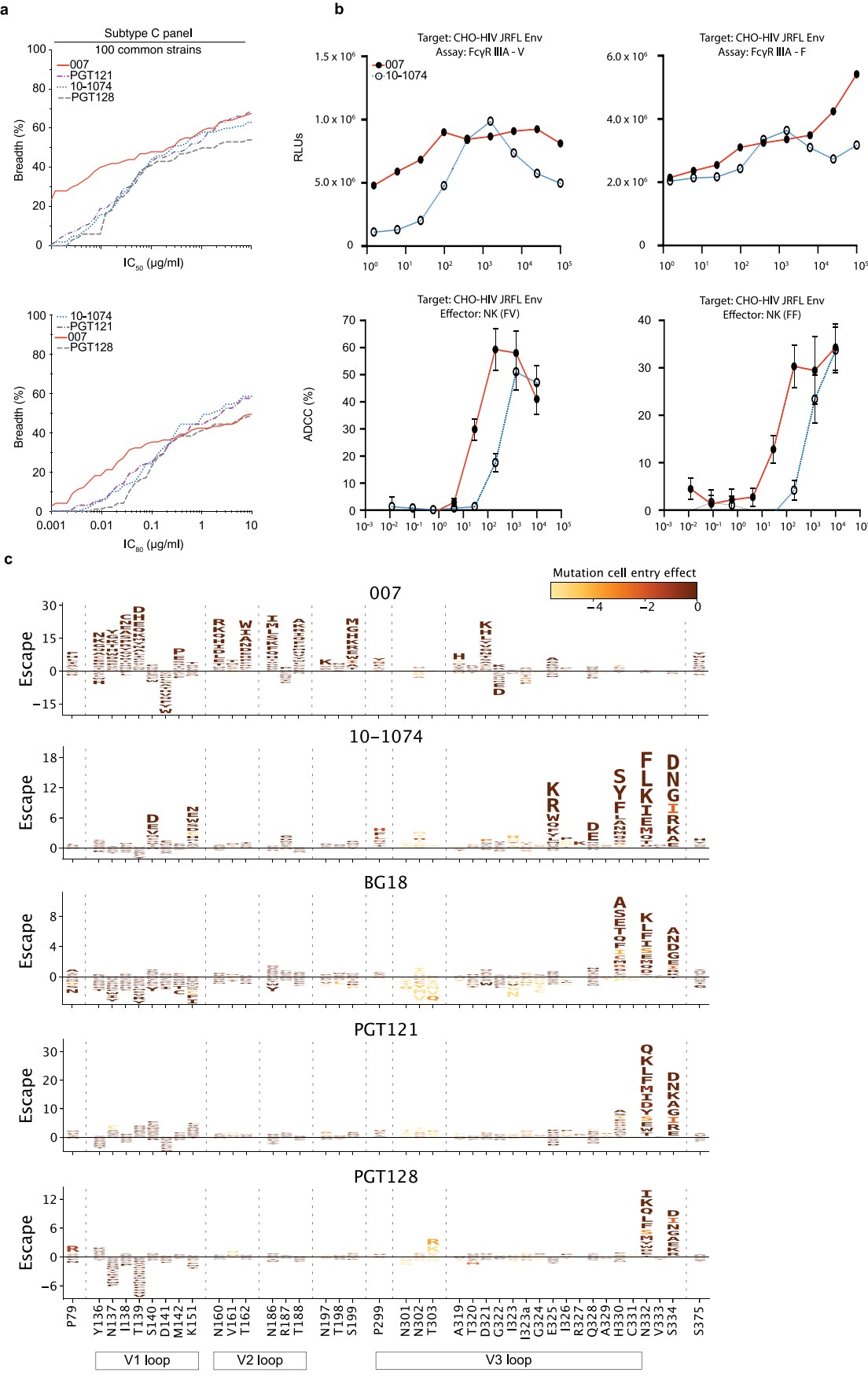

**Extended Data Fig. 8 | See next page for caption.**

**Extended Data Fig. 8 | Antiviral activity against the 100-strain Subtype C panel, effector function against Env expressing cells and HIV-1 ENV$_{BF520}$ deep mutational escape map. a**, Neutralization breadth (%) and potency (IC$_{50}$/IC$_{80}$) of 007 against the 100-strain Subtype C pseudovirus panel. Breadth (%) was calculated using a cutoff of ≤10 μg/mL. Data are shown for identical virus strains across each panel with available reference neutralization data sourced from CATNAP database. **b**, 007 demonstrates potent ADCC against HIV envelope expressing cells. FcγRIIIa signaling was assessed using an ADCC reporter bioassay (RLUs) and CHO cell overexpressing HIV Env as targets. Jurkat reporter cells expressed either FcγRIIIa-F or FcγRIIIa-V alleles (representative data, n = 2). V158 is the high affinity allele for FcγRIIIa and F158 is the low-affinity allele. ADCC target cell killing (%) was assessed using NK cells as effectors and CHO-HIV Env as target cells (representative data of n = 3 experiments; mean ± SD of duplicate points). **c**, Logo plots showing effects of mutations on neutralization escape in the HIV Env BF520 strain for antibodies 007, 10-1074, BG18, PGT121, and PGT128. The height of each letter represents the effect of that amino acid mutation on antibody neutralization, with positive heights (letters above the zero line) indicating mutations that cause escape, and negative heights (letters below the zero line) indicating mutations that increase neutralization. Letters are colored by the effect of that mutation on Env-mediated cell entry function, with yellow corresponding to reduced cell entry and brown corresponding to neutral effects on cell entry. Only key sites are shown. See https://dms-vep.org/HIV_Envelope_BF520_DMS_007/htmls/all_antibodies_and_cell_entry_overlaid.html for interactive versions of the escape maps that show all mutations. The deep mutational scanning measured effects of mutations on escape from antibody 10-1074 shown in this figure are previously published[41].

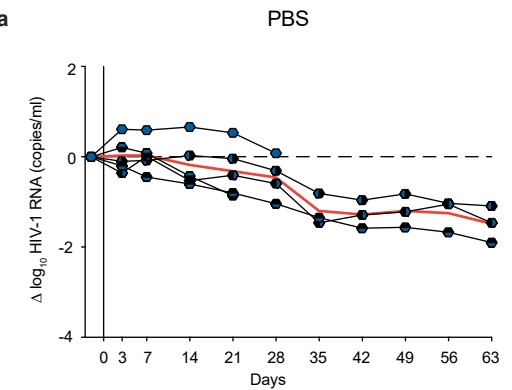
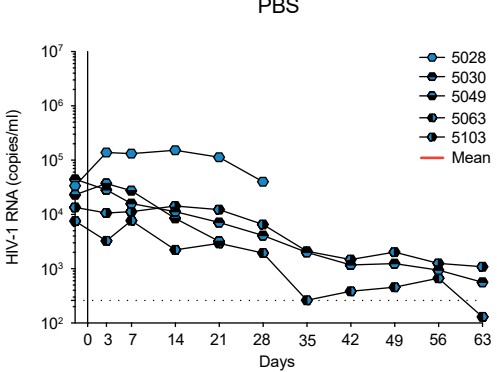

**Extended Data Fig. 9 | Viral loads of PBS-treated mice. a**, Viral loads in HIV-1$_{ADA}$-infected humanized mice from the PBS-treated control group. The graphs depict relative log$_{10}$ changes from baseline viral loads (left) and absolute HIV-1 RNA plasma copies/mL (right) under PBS administrations. Dashed lines in the right graph represent the lower limit of quantification (LLQ) of the qPCR assay (260 copies/mL). Red lines indicate the average log$_{10}$ changes relative to baseline viral loads (day −2).

# Reporting Summary

## Statistics

For all statistical analyses, confirm that the following items are present in the figure legend, table legend, main text, or Methods section.

| n/a | Confirmed | |
|---|---|---|
| ☐ | ☒ | The exact sample size (*n*) for each experimental group/condition, given as a discrete number and unit of measurement |
| ☐ | ☒ | A statement on whether measurements were taken from distinct samples or whether the same sample was measured repeatedly |
| ☐ | ☒ | The statistical test(s) used AND whether they are one- or two-sided *Only common tests should be described solely by name; describe more complex techniques in the Methods section.* |
| ☒ | ☐ | A description of all covariates tested |
| ☒ | ☐ | A description of any assumptions or corrections, such as tests of normality and adjustment for multiple comparisons |
| ☐ | ☒ | A full description of the statistical parameters including central tendency (e.g. means) or other basic estimates (e.g. regression coefficient) AND variation (e.g. standard deviation) or associated estimates of uncertainty (e.g. confidence intervals) |
| ☐ | ☒ | For null hypothesis testing, the test statistic (e.g. *F*, *t*, *r*) with confidence intervals, effect sizes, degrees of freedom and *P* value noted *Give P values as exact values whenever suitable.* |
| ☒ | ☐ | For Bayesian analysis, information on the choice of priors and Markov chain Monte Carlo settings |
| ☒ | ☐ | For hierarchical and complex designs, identification of the appropriate level for tests and full reporting of outcomes |
| ☐ | ☒ | Estimates of effect sizes (e.g. Cohen's *d*, Pearson's *r*), indicating how they were calculated |

*Our web collection on statistics for biologists contains articles on many of the points above.*

## Software and code

Policy information about availability of computer code

| Data collection | GraphPad Prism V.8: Illustration of data and statistical parameter. Statistical analysis of data.<br>Microsoft Excel V16.16.27 (201012): Illustration of data and statistical parameter. Statistical analysis of data<br>BD FACSDiva 6.1.2: Generation of flow cytometry data.<br>ELISA measurements were performed using Tecan's Sunrise absorbance microplate reader and associated software.<br>Neutralizing activity and luminescence were measures using BertholdTech Tristar2S and associated device software<br>B cell repertoire sequence data was generated by an unbiased template-switch-based NGS approach (Ehrhardt et al., Nature Medicine 2019)<br>Sanger sequencing of antibody v genes were gained by Eurofins genomics.<br>Clinical data for some study individuals was documented in an electronic case report form (eCRF) using the online cohort platform<br>ClinicalSurveys.net hosted by QuestBack, Oslo, Norway on servers of UHC, Cologne, Germany, as part of a software-as-a-service agreement.<br>All computer code and data for the deep mutational scanning is publicly available on GitHub (https://dms-vep.org/<br>HIV_Envelope_TRO11_DMS_007/htmls/all_antibodies_and_cell_entry_overlaid.html). |
|---|---|
| Data analysis | Data, quantifications and statistical analyses were performed using GraphPad Prism V.8, Microsoft Excel V16.16.27 (201012), BD FlowJo V.10, and Geneious Prime (v.2020.2.4). Raw read pre-processing of B cell NGS data was performed with an in-house pipeline primarily based on self-written Python scripts, IgBLAST, Clustal Omega, and the pRESTO toolkit. Raw IGHV and immunoglobulin light chain variable (IGLV) sequences derived from NGS and single B cell sequencing data were annotated to human V, D, and J germline reference sequences using MiXCR software. All computer code and data for the deep mutational scanning is publicly available on GitHub (https://dms-vep.org/ HIV_Envelope_TRO11_DMS_007/htmls/all_antibodies_and_cell_entry_overlaid.html). |

For manuscripts utilizing custom algorithms or software that are central to the research but not yet described in published literature, software must be made available to editors and reviewers. We strongly encourage code deposition in a community repository (e.g. GitHub). See the Nature Portfolio guidelines for submitting code & software for further information.

# Data

Policy information about <u>availability of data</u>

All manuscripts must include a <u>data availability statement</u>. This statement should provide the following information, where applicable:

- Accession codes, unique identifiers, or web links for publicly available datasets
- A description of any restrictions on data availability
- For clinical datasets or third party data, please ensure that the statement adheres to our <u>policy</u>

Nucleotide sequences of bnAb 007 and clonal members have been deposited at GenBank under accession codes XXX. The NGS B cell repertoire data analysed in this study have been deposited in the Sequence Read Archive (SRA) under accession codes SAMN29624595 to SAMN29624713 [https://www.ncbi.nlm.nih.gov/sra?linkname=bioproject_sra_all&from_uid=857338] and the BioProject database under accession code PRJNA857338. Cryo-EM maps and models have been deposited in the Electron Microscopy Data Bank (EMDB) and Protein Data Bank with accession codes: EMD-70018 and PDB ID 9O2Q (007-BG505v2, 0 Fabs bound), EMD-70019 and PDB ID 9O2R (007-BG505v2, 1 Fab bound), EMD-70020 and PDB ID 9O2S (007-BG505v2, 2 Fabs bound), EMD-70021 and PDB ID 9O2T (007-BG505v2, 3 Fabs bound), EMD-70022 and PDB ID 9O2U (007-BG505v2, IgG crosslinked trimer-dimer). Clinical data are available upon request to the corresponding author (F.K.) provided that there is no reasonable risk of deanonymizing study participants and/or may require a Material Transfer Agreement (MTA). Individual donor data cannot be shared due to privacy restrictions. Requests will be responded to within 2 weeks.

# Research involving human participants, their data, or biological material

Policy information about studies with <u>human participants or human data</u>. See also policy information about <u>sex, gender (identity/presentation), and sexual orientation</u> and <u>race, ethnicity and racism</u>.

| | |
|---|---|
| Reporting on sex and gender | In this study, we collected data about the sex of one study individuals. No data was collected on gender. Therefore, we only use the term "sex" in our manuscript. We present data from one female study participants. Samples were collected irrespective of sex or gender. Sex or gender was not relevant to the study design. |
| Reporting on race, ethnicity, or other socially relevant groupings | The origin of samples was Mbeya, Tanzania. |
| Population characteristics | Population characteristic about sex of the described study individuals are as following: <br> 1.) Female sex: 100% |
| Recruitment | The donor was recruited from a hospital in Tanzania (Mbeya). Recruitment occurred during routine visits to this facility. Enrollment was unbiased as it was determined by the treating physician without prior selection or self-selection by the patients. However, the physician's choice could introduce potential bias in patient selection. While we recognize this possible source of bias in our recruitment approach, we are confident that it did not influence the study outcomes. |
| Ethics oversight | Large-blood draw samples were collected according to protocols reviewed and approved by the Institutional Review Board (IRB) of the University of Cologne (study protocols 13-364 and 16-054) and local IRBs. Compensation was provided in line with institutional and ethical guidelines to reimburse time and expenses without exerting undue influence. |

Note that full information on the approval of the study protocol must also be provided in the manuscript.

# Field-specific reporting

Please select the one below that is the best fit for your research. If you are not sure, read the appropriate sections before making your selection.

☒ Life sciences ☐ Behavioural & social sciences ☐ Ecological, evolutionary & environmental sciences

For a reference copy of the document with all sections, see <u>nature.com/documents/nr-reporting-summary-flat.pdf</u>

# Life sciences study design

All studies must disclose on these points even when the disclosure is negative.

| | |
|---|---|
| Sample size | No formal sample size calculation was performed. The study aimed to identify potent broadly neutralizing antibodies from a single HIV-1 elite neutralizer by analyzing all available HIV-1–reactive B cell clones isolated from this donor. Sample size was therefore determined by biological availability rather than statistical power considerations. The total number of clones analyzed was sufficient to comprehensively investigate all isolated HIV-1-reactive B cell clones. |
| Data exclusions | For NGS data, reads were initially filtered for a mean Phred score of 25 and read-lengths of at least 250 bp. Consensus sequences (based on UMIs) were excluded, when the corresponding UMI was found less than 3 times. For sequence analysis of single cell-derived v gene sequences, chromatograms were first filtered based on a minimum Phred score of 28 and a length of at least 240 nucleotides. Sequences were then processed using IgBLAST to annotate them, and trimmed to extract only the variable region, extending from FWR1 to the end of the J gene. Base calls within the variable region with a Phred score below 16 were masked. Sequences containing over 15 masked nucleotides, stop codons, or frameshift mutations were excluded from subsequent analysis. No other data were excluded from analyses. <br><br> All anti-HIV-1 bNAbs reference antibodies were functionally validated through neutralization assays against the global HIV-1 pseudovirus |

panel. The resulting IC$_{50}$ and IC$_{80}$ values for each antibody were compared to historical data available in the CATNAP database. Only those antibodies exhibiting less than a threefold deviation in IC$_{50}$/IC$_{80}$ values relative to the reference data were included in subsequent functional analyses.

| | |
|---|---|
| Replication | For antibody screening assays, the neutralizing activity of monoclonal antibodies was first assessed without duplicates. Determination of the neutralizing activity of antibodies against other virus panels were conducted in duplicates and the geometric mean was reported. All attempts at replication were successful. |
| Randomization | Randomization was relevant for the in vivo treatment experiments; mice were randomized into treatment or control groups based on viral load, age, and time since humanization to ensure even distribution across groups. |
| Blinding | Researches conducting experiments were aware of the study ID for each individual they handled, but they were blinded to the participants' clinical data. The aim was to identify and characterize a broadly neutralizing antibody (bNAb) from a single HIV-1 elite neutralizer. All experiments were exploratory and descriptive in nature. Data collection and analysis involved quantitative, assay-based measurements (e.g., neutralization assays, binding analyses, structural studies) that were not influenced by investigator bias. Therefore, blinding was not applicable to the experimental design. |

# Reporting for specific materials, systems and methods

We require information from authors about some types of materials, experimental systems and methods used in many studies. Here, indicate whether each material, system or method listed is relevant to your study. If you are not sure if a list item applies to your research, read the appropriate section before selecting a response.

### Materials & experimental systems

| n/a | Involved in the study |
|---|---|
| ☐ | ☒ Antibodies |
| ☐ | ☒ Eukaryotic cell lines |
| ☒ | ☐ Palaeontology and archaeology |
| ☐ | ☒ Animals and other organisms |
| ☐ | ☒ Clinical data |
| ☒ | ☐ Dual use research of concern |
| ☒ | ☐ Plants |

### Methods

| n/a | Involved in the study |
|---|---|
| ☒ | ☐ ChIP-seq |
| ☐ | ☒ Flow cytometry |
| ☒ | ☐ MRI-based neuroimaging |

## Antibodies

| | |
|---|---|
| Antibodies used | Anti-human CD20-AF700 (clone 2H7), BD Bioscience, Cat#560631, RRID: AB_2687799, dilution 1:80<br>Anti-human CD19-AF700 (clone HIB19), BD Bioscience, Cat#557921, RRID: AB_396942, dilution 1:80<br>Anti-human IgG-APC (clone G18-145), BD Bioscience, Cat#550931, RRID AB_2738854, dilution 1:20<br>Anti-human CD27-PE (Clone M-T271), BD Bioscience, CAT#560985, RRID: AB_10563213, dilution 1:20<br>Peroxidase AffiniPure Goat Anti-human IgG, Fcy fragment specific, Jackson ImmunoResearch, Cat#109-035-098, RRID: AB_237586<br>IgG1 kappa from human myeloma plasma (Sigma Aldrich, cat. no. I5154), dilution 1:5000<br>Monoclonal anti-HIV-1 bNAb 10-1074, sequence retrieved from CATNAP under record #2777, starting concentration of neutralization assays of 10, 25 or 50 µg/ml<br>Monoclonal anti-HIV-1 bNAb BG18, sequence retrieved from CATNAP under record #3623, starting concentration of neutralization assays of 10, 25 or 50 µg/ml<br>Monoclonal anti-HIV-1 bNAb PGT121, Genbank ID heavy chain: JN201894, light chain: JN201911, starting concentration of neutralization assays of 10, 25 or 50 µg/ml<br>Monoclonal anti-HIV-1 bNAb PGT128, Genbank ID heavy chain: JN201900, light chain: JN201917, starting concentration of neutralization assays of 10, 25 or 50 µg/ml |
| Validation | All antibodies from BD bioscience and Jackson ImmunoResearch were controlled for human reactivity during manufcaturer's quality control. All anti-HIV-1 bNAbs were tested against the global pseudovirus panel to guarantee stability and functionality. All anti-HIV-1 bNAbs reference antibodies were functionally validated through neutralization assays against the global HIV-1 pseudovirus panel. The resulting IC$_{50}$ and IC$_{80}$ values for each antibody were compared to historical data available in the CATNAP database. Only those antibodies exhibiting less than a threefold deviation in IC$_{50}$/IC$_{80}$ values relative to the reference data were included in subsequent functional analyses. |

## Eukaryotic cell lines

Policy information about cell lines and Sex and Gender in Research

| | |
|---|---|
| Cell line source(s) | TZM-bl cells (NIH AIDS Reagent Program, Ref# ARP5011), HEK293T (American Type Culture Collection, Ref# CRL-3216), HEK293-6E cells (National Research Council of Canada, Cat. No. NR16-179/2017E). The sex of all cells was female. |

| Authentication | Cell line were authenticated with STR analysis. |
|---|---|
| Mycoplasma contamination | Cell line was tested negative for mycoplasma contamination. |
| Commonly misidentified lines (See ICLAC register) | No commonly misidentified cell lines were used in the study. |

# Animals and other research organisms

Policy information about studies involving animals; ARRIVE guidelines recommended for reporting animal research, and Sex and Gender in Research

| Laboratory animals | NOD.Cg-Rag1tm1momIl2rgtm1Wjl/SzJ (NRG) mice were obtained from The Jackson Laboratory<br>B6.Cg-Fcgrttm1Dcr Prkdcscid Tg(FCGRT)32Dcr/DcrJ (FcRn) mice were obtained from The Jackson Laboratory |
|---|---|
| Wild animals | No wild animals were used for this study |
| Reporting on sex | For treatment experiments using NOD.Cg-Rag1tm1momIl2rgtm1Wjl/SzJ (NRG) mice obtained from The Jackson Laboratory:<br>14 male mice, 19 female mice<br><br>For determination of antibody PKs in vivo using B6.Cg-Fcgrttm1Dcr Prkdcscid Tg(FCGRT)32Dcr/DcrJ (FcRn) mice obtained from The Jackson Laboratory:<br>12 female mice (007, 10-1074, 3BNC117) |
| Field-collected samples | No field samples were collected in this study. |
| Ethics oversight | All procedures involving mice in this study were sanctioned by the State Agency for Nature, Environmental Protection, and Consumer Protection North Rhine-Westphalia (LANUV). |

Note that full information on the approval of the study protocol must also be provided in the manuscript.

# Clinical data

Policy information about clinical studies
All manuscripts should comply with the ICMJE guidelines for publication of clinical research and a completed CONSORT checklist must be included with all submissions.

| Clinical trial registration | Studies have not been registered |
|---|---|
| Study protocol | Study protocols will be made available on request from the corresponding author. |
| Data collection | Data collection was carried out at each site using either local data monitoring systems or the centralized online cohort platform, ClinicalSurveys.net. The collected data were then harmonized and consolidated by staff at the University Hospital Cologne. |
| Outcomes | Non-interventional study. No outcome parameters were defined. |

# Plants

| Seed stocks | No plants were used in this study |
|---|---|
| Novel plant genotypes | No plants were used in this study |
| Authentication | No plants were used in this study |

# Flow Cytometry

## Plots

Confirm that:

☒ The axis labels state the marker and fluorochrome used (e.g. CD4-FITC).

☐ The axis scales are clearly visible. Include numbers along axes only for bottom left plot of group (a 'group' is an analysis of identical markers).

☒ All plots are contour plots with outliers or pseudocolor plots.

☒ A numerical value for number of cells or percentage (with statistics) is provided.

## Methodology

| | |
|---|---|
| Sample preparation | PBMCs of study individuals were purified by standard density gradient centrifugation using Histopaque (Sigma Aldrich) and LeucoSep tubes (Greiner Bio-one). Cells were sorted at -150°C in 90% (v/v) FBS (Sigma Aldrich) and 10% (v/v) DMSO (Sigma Aldrich). For single B cell sorts, cells were stained with anti-human CD19-AF700 (BD Bioscience), anti-human IgG-APC (BD Bioscience), DAPI (Thermo Fisher), and the HIV-1 env bait BG505.SOSIP.664-GFP for 30 min on ice. Env-reactive CD19+IgG +DAPI- single cells were sorted into 96-well plates containing 4 µl of lysis buffer. |
| Instrument | FACSAria Fusion (Becton Dickinson) |
| Software | BD FACSDIVA, FlowJo10 |
| Cell population abundance | Post-sort fractions were not re-analyzed. |
| Gating strategy | Gating on lymphocytes, live cells, CD19+ and IgG+ and the GFP+ (BG505.SOSIP.664-GFP) for HIV-1 reactive cells. |

☒ Tick this box to confirm that a figure exemplifying the gating strategy is provided in the Supplementary Information.

