## [Peer Review File · Nature Immunology]

Identification of a potent V3 glycan site broadly neutralizing antibody targeting an N332 glycan-independent epitope

Corresponding Author: Professor Florian Klein

Version 0:

Reviewer comments:

Reviewer #1

(Remarks to the Author)

Gieselmann et al describe the identification of a potent broadly neutralizing anti-HIV Env mAb targeting the V3-glycan supersite in a glycan-independent way. In a large screening, the authors identify an Exceptional (in terms of magnitude and neutralization fingerprint) Elite Neutralizer individual. Trimeric-Env specific B cells were sorting identifies the 007 mAb as a broadly neutralizing antibody with a mechanism of action similar to EPTC112 (N322 glycan independent) but more potent and broad. The authors characterize, the stoichiometry of binding, the impact of avidity, the complementarity with other anti-V3 site antibodies, the landscape of resistance mutations and the in vivo activity. Altogether, the work is relevant for HIV vaccine, prevention and cure research.

General comments:

1- The mode of action of 007. In a first attempt to define the interaction of 007 with Env, the authors perform CryoEM analysis of Fab fragments of the antibody. After realizing the strong role of avidity in neutralizing activity, they also explore the binding of the full antibody. The authors conclude that given the distance of antibody arms and intra Spike epitopes, the most likely mechanism of action is viral aggregation. However, no strong conclusion could be reached. Two comments in this regard:

- The authors rule out the intraspine binding considering the IgG1 backbone in which the variable regions were cloned. However, the natural antibody could be, for instance an IgG3, with longer hinge and flexibility. Could an IgG3 antibody be consistent with intraspine binding? Could the authors explore the subclass of the original antibody from the RNA/retrotranscribed DNA or first amplicons obtained from the 007 B-cell clone?
- Analyzing aggregation of viral particles induced by 007 antibody, even semi-quantitatively, by EM or any other technique would be helpful to define the final mechanism and support the conclusion.

2- The therapeutic potential. Two comments also about that:

- The pK decay in human FcRn transgenic mice (Supp Fig 3) is faster than standard antibodies in clinical trials (10-1074 and 3BNC117). Could this impact also PK profile in the mice used in vivo experiments (huFcRn-). Did the authors analyze PK in the latter animal model to define the administration regimen?
- Going back to the combination 10-1074 and 3BNC117, which has been tested in NHP and humans and seems a good strategy combining anti CD4bs (medium potency/high coverage) and anti V3 (more potent but with lower coverage). However, 007 directly/indirectly competes with 3BNC117 and 1-18 (Figure 1c), could this observation limit the combination of 007 with anti CD4bs antibodies?

3- The in vivo activity. Although the sample size is limited, monotherapy with 007 seems to lead to a faster escape compared to 10-1074 (Figure 6a). As only amino acid mutations are provided, it is difficult to evaluate the number of real mutations required for generating resistance to each antibody. Could the authors comment on a possible difference in genetic barrier?

Specific comments:

- Figure 1c. It is unclear why the self-control does not show a clear inhibition, as other antibodies do.
- Figure 2b. Since the 007 closest antibody (in terms of binding) is EPTC112, it should be included in this figure. PDB should be available from the original publication
- Figure 2i. Could cryoEM data help to understand the role of glycans in the binding. How many mannose molecules could be resolved in CryoEM in bound and unbound forms?
- IN Figure 4, the authors highlight that 10-1074 resistant viruses are mostly sensitive to 007 (60% of them). However, the

separate resistance mechanism shown in Figure 5 and in vivo data shown in Figure 6, suggest that a similar situation could be observed for the sensitivity to 10-1074 of 007 resistant viruses. Adding such an analysis to Figure 4 could give a more balanced view of the complementarity of both antibodies.

Minor comments:

8- Data in extended Figure 9 panel b, using reporter Jurkat cells (top) shows unclear differences, as compared to the ADCC measures (bottom). For instance, in a Log scale, no relevant differences between 007 and 10-1074 antibodies in FcR1IIa F variant can be observed

9- For a better comparison, of the effect of the antibodies the mean of the HIV RNA or $\Delta\log$ 10 HIV RNA of the PBS control group (shown in Supp Fig 10) should be added to Figure 6

Reviewer #2

(Remarks to the Author)

Gieselmann et al. identify and characterize antibody '007', which has potent neutralization (0.012 $\mu\text{g}/\text{mL}$) and decent breadth (~70% on a 217-strain panel). In specific, they sort and sequence B cells from donor EN01, the top neutralizer from a Tanzania cohort of over 2,000 individuals living with HIV for recognition of BG505 Env trimer, identifying 23 B cell lineages and producing and assessing 48 monoclonal antibodies by neutralization on a cross-clade panel of 6 viruses, identifying antibody 007, with broad and potent recognition.

Cryo-EM structure determination of Fab 007 with BG505 DS-SOSIP, revealed recognition similar to antibodies of the V3 glycan supersite, though with recognition of N156 and N301, though not of N332, as well as of the V3 motif - 324GD/NIR327. Recognition by 007 was most similar to EPTC112, a recently described V3 glycan antibody, which, like 007, does not need N332 for neutralization. Neutralization by 007 IgG was much higher than for Fab, though the exact reason for this was unclear, as the structure of 007 to the prefusion-closed conformation showed angles of approach incompatible with bivalent binding to a single trimer. Despite 007 recognizing the V3 glycan supersite, its neutralization was complementary to other V3 glycan supersite antibodies, and it had a distinct escape profile (revealed by deep mutational scanning). In vivo, 007 suppressed viremia in humanized mice, with dual targeting by 007 and 10-1074 delaying viral rebound.

Overall, the authors generally do a nice job describing the identification and properties of antibody 007.

One area where they are a bit lax is a precise description of the neutralization dependence on glycans 156, 301, and 332/334. It would be clearest for site-specific glycan mutants to be assessed for impact on neutralization; alternatively, such information can be extracted by analysis of sequences of viral strains accessed for 007 neutralization sensitivity. It would be helpful to define the 007 sensitivity/resistance of isolates that have or do not have glycans at 156, 301, and/or 332/334. (When showing this information, please also show comparitors including EN01 and PGT128, to provide readers with insight into how different 007 is from prior identified V3 glycan antibodies.)

While the authors highlight donor EN01, they do not analyze this neutralization in the context of the identified 007 lineage. It would be helpful for the authors to provide feedback on this top elite neutralizer by calculating and reporting the extent that the neutralization of donor EN01 can be explained by the neutralization of antibody 007 – as well as the concentration of antibody 007 needed to best account for serum neutralization.

The authors identify a fascinating difference in Fab versus IgG neutralization. To provide data separating potential explanations for this difference, the authors should measure actual affinities, such as Fab affinity and apparent IgG affinity of 007 to prefusion-closed Env trimer stabilized by DS as well as only SOSIP stabilized to see how conformational flexibility impacts neutralization.

Also, it's interesting that 007 binding overlaps that of other V3 glycan antibodies, yet its neutralization fingerprint as well as escape is so different. The authors, unfortunately, do not quantify this difference in a general way. It would be helpful in this regard for the authors to compare epitope similarity versus neutralization fingerprint to see how much of an outlier 007 is versus other antibodies. I suggest using variable domain overlap after superposition on the prefusion-closed Env trimer as one potential metric for epitope-overlap on the (X-axis) versus dendrogram length of the neutralization profile (Fig. 1e). If such an analysis were done for all catnip-deposited antibodies with structures that can be superimposed on the prefusion closed-trimer, it would provide quantitative insight into how different 007 is versus other HIV-1 bNAb antibodies.

Other modifications:

In Fig. 2b, please show a matrix of % variable domain overlap, to complement the graphic provided.

Decision Letter:

1st Jul 2025

Dear Professor Klein,

Your Article, "Identification of a broad and potent V3 glycan site bNAbs targeting an N332 glycan-independent epitope" has now been seen by 2 referees. You will see from their comments copied below that while they find your work of considerable potential interest, they have raised quite substantial concerns that must be addressed. In light of these comments, we cannot accept the manuscript for publication, but would be very interested in considering a revised version that addresses these serious concerns.

We hope you will find the referees' comments useful as you decide how to proceed. If you wish to submit a substantially revised manuscript, please bear in mind that we will be reluctant to approach the referees again in the absence of major revisions.

In particular, reviewer #1 would like you to do further experimentation regarding the mechanism of action of 007, the therapeutic potential and the in vivo activity, as well as do further analysis of resistant viruses and assess the role of glycans in the binding. Reviewer #2 thinks that the precise description of the neutralization dependence on glycans 156, 301, and 332/334 should be improved, you should analyze this neutralization in the context of the identified 007 lineage, measure actual affinities, and quantify the different neutralization fingerprint versus epitope.

If you choose to revise your manuscript taking into account all reviewer and editor comments, please highlight all changes in the manuscript text file.

* If you have not done so already please begin to revise your manuscript so that it conforms to our Article format instructions at <http://www.nature.com/ni/authors/index.html>. Refer also to any guidelines provided in this letter.

The Reporting Summary can be found here:

When submitting the revised version of your manuscript, please pay close attention to our [href="https://www.nature.com/nature-portfolio/editorial-policies/image-integrity">Digital Image Integrity Guidelines.](https://www.nature.com/nature-portfolio/editorial-policies/image-integrity) and to the following points below:

Extended Data figures and tables are online-only (appearing in the online PDF and full-text HTML version of the paper), peer-reviewed display items that provide essential background to the Article but are not included in the printed version of the paper due to space constraints or being of interest only to a few specialists. A maximum of ten Extended Data display items (figures and tables) is typically permitted. When re-submitting your manuscript, please ensure that any supplementary figures and tables that are more critical to the manuscript's conclusions are converted to Extended data to increase these data's visibility.

Link Redacted

If you wish to submit a suitably revised manuscript we would hope to receive it within 6 months. If you cannot send it within this time, please let us know. We will be happy to consider your revision so long as nothing similar has been accepted for publication at Nature Immunology or published elsewhere.

Nature Immunology is committed to improving transparency in authorship. As part of our efforts in this direction, we are now requesting that all authors identified as 'corresponding author' on published papers create and link their Open Researcher and Contributor Identifier (ORCID) with their account on the Manuscript Tracking System (MTS), prior to acceptance. ORCID helps the scientific community achieve unambiguous attribution of all scholarly contributions. You can create and link your ORCID from the home page of the MTS by clicking on 'Modify my Springer Nature account'. For more information please visit www.springernature.com/orcid.

Thank you for the opportunity to review your work.

Sincerely,
Paula

Paula Jauregui, PhD
Senior Editor
Nature Immunology

Referee expertise:

Referee #1: HIV antibodies and HIV env

Referee #2: CryoEM and HIV vaccine and NHP

Reviewers' Comments:

Reviewer #1 (Remarks to the Author):

Gieselmann et al describe the identification of a potent broadly neutralizing anti-HIV Env mAb targeting the V3-glycan supersite in a glycan-independent way. In a large screening, the authors identify an Exceptional (in terms of magnitude and neutralization fingerprint) Elite Neutralizer individual. Trimeric-Env specific B cells were sorting identifies the 007 mAb as a broadly neutralizing antibody with a mechanism of action similar to EPTC112 (N322 glycan independent) but more potent and broad. The authors characterize, the stoichiometry of binding, the impact of avidity, the complementarity with other anti-V3 site antibodies, the landscape of resistance mutations and the in vivo activity. Altogether, the work is relevant for HIV vaccine, prevention and cure research.

General comments:

1- The mode of action of 007. In a first attempt to define the interaction of 007 with Env, the authors perform CryoEM analysis of Fab fragments of the antibody. After realizing the strong role of avidity in neutralizing activity, they also explore the binding of the full antibody. The authors conclude that given the distance of antibody arms and intra Spike epitopes, the most likely mechanism of action is viral aggregation. However, no strong conclusion could be reached. Two comments in this regard:

a. The authors rule out the intraspine binding considering the IgG1 backbone in which the variable regions were cloned. However, the natural antibody could be, for instance an IgG3, with longer hinge and flexibility. Could an IgG3 antibody be consistent with intraspine binding? Could the authors explore the subclass of the original antibody from the RNA/retrotranscribed DNA or first amplicons obtained from the 007 B-cell clone?

b. Analyzing aggregation of viral particles induced by 007 antibody, even semi-quantitatively, by EM or any other technique would be helpful to define the final mechanism and support the conclusion.

2- The therapeutic potential. Two comments also about that:

a. The pK decay in human FcRn transgenic mice (Supp Fig 3) is faster than standard antibodies in clinical trials (10-1074 and 3BNC117). Could this impact also PK profile in the mice used in vivo experiments (huFcRn-). Did the authors analyze PK in the latter animal model to define the administration regimen?

b. Going back to the combination 10-1074 and 3BNC117, which has been tested in NHP and humans and seems a good strategy combining anti CD4bs (medium potency/high coverage) and anti V3 (more potent but with lower coverage). However, 007 directly/indirectly competes with 3BNC117 and 1-18 (Figure 1c), could this observation limit the combination of 007 with anti CD4bs antibodies?

3- The in vivo activity. Although the sample size is limited, monotherapy with 007 seems to lead to a faster escape compared to 10-1074 (Figure 6a). As only amino acid mutations are provided, it is difficult to evaluate the number of real mutations required for generating resistance to each antibody. Could the authors comment on a possible difference in genetic barrier?

Specific comments:

- 4- Figure 1c. It is unclear why the self-control does not show a clear inhibition, as other antibodies do.
- 5- Figure 2b. Since the 007 closest antibody (in terms of binding) is EPTC112, it should be included in this figure. PDB should be available from the original publication
- 6- Figure 2i. Could cryoEM data help to understand the role of glycans in the binding. How many mannose molecules could be resolved in CryoEM in bound and unbound forms?
- 7- IN Figure 4, the authors highlight that 101-1074 resistant viruses are mostly sensitive to 007 (60% of them). However, the separate resistance mechanism shown in Figure 5 and in vivo data shown in Figure 6, suggest that a similar situation could be observed for the sensitivity to 10-1074 of 007 resistant viruses. Adding such an analysis to Figure 4 could give a more balanced view of the complementarity of both antibodies.

Minor comments:

- 8- Data in extended Figure 9 panel b, using reporter Jurkat cells (top) shows unclear differences, as compared to the ADCC measures (bottom). For instance, in a Log scale, no relevant differences between 007 and 10-1074 antibodies in FcR11a F variant can be observed
- 9- For a better comparison, of the effect of the antibodies the mean of the HIV RNA or $\Delta\log_{10}$ HIV RNA of the PBS control group (shown in Supp Gig 10) should be added to Figure 6

Reviewer #2 (Remarks to the Author):

Gieselmann et al. identify and characterize antibody '007', which has potent neutralization (0.012 ug/mL) and decent breadth (~70% on a 217-strain panel). In specific, they sort and sequence B cells from donor EN01, the top neutralizer from a Tanzania cohort of over 2,000 individuals living with HIV for recognition of BG505 Env trimer, identifying 23 B cell lineages and producing and assessing 48 monoclonal antibodies by neutralization on a cross-clade panel of 6 viruses, identifying antibody 007, with broad and potent recognition.

Cryo-EM structure determination of Fab 007 with BG505 DS-SOSIP, revealed recognition similar to antibodies of the V3 glycan supersite, though with recognition of N156 and N301, though not of N332, as well as of the V3 motif - 324GD/NIR327. Recognition by 007 was most similar to EPTC112, a recently described V3 glycan antibody, which, like 007, does not need N332 for neutralization. Neutralization by 007 IgG was much higher than for Fab, though the exact reason for this was unclear, as the structure of 007 to the prefusion-closed conformation showed angles of approach incompatible with bivalent binding to a single trimer. Despite 007 recognizing the V3 glycan supersite, its neutralization was complementary to other V3 glycan supersite antibodies, and it had a distinct escape profile (revealed by deep mutational scanning). In vivo, 007 suppressed viremia in humanized mice, with dual targeting by 007 and 10-1074 delaying viral rebound.

Overall, the authors generally do a nice job describing the identification and properties of antibody 007.

One area where they are a bit lax is a precise description of the neutralization dependence on glycans 156, 301, and 332/334. It would be clearest for site-specific glycan mutants to be assessed for impact on neutralization; alternatively, such information can be extracted by analysis of sequences of viral strains accessed for 007 neutralization sensitivity. It would be helpful to define the 007 sensitivity/resistance of isolates that have or do not have glycans at 156, 301, and/or 332/334. (When showing this information, please also show comparitors including EN01 and PGT128, to provide readers with insight into how different 007 is from prior identified V3 glycan antibodies.)

While the authors highlight donor EN01, they do not analyze this neutralization in the context of the identified 007 lineage. It would be helpful for the authors to provide feedback on this top elite neutralizer by calculating and reporting the extent that the neutralization of donor EN01 can be explained by the neutralization of antibody 007 – as well as the concentration of antibody 007 needed to best account for serum neutralization.

The authors identify a fascinating difference in Fab versus IgG neutralization. To provide data separating potential explanations for this difference, the authors should measure actual affinities, such as Fab affinity and apparent IgG affinity of 007 to prefusion-closed Env trimer stabilized by DS as well as only SOSIP stabilized to see how conformational flexibility impacts neutralization.

Also, it's interesting that 007 binding overlaps that of other V3 glycan antibodies, yet its neutralization fingerprint as well as escape is so different. The authors, unfortunately, do not quantify this difference in a general way. It would be helpful in this regard for the authors to compare epitope similarity versus neutralization fingerprint to see how much of an outlier 007 is versus other antibodies. I suggest using variable domain overlap after superposition on the prefusion-closed Env trimer as one potential metric for epitope-overlap on the (X-axis) versus dendrogram length of the neutralization profile (Fig. 1e). If such an analysis were done for all catnip-deposited antibodies with structures that can be superimposed on the prefusion closed-trimer, it would provide quantitative insight into how different 007 is versus other HIV-1 bNAb antibodies.

Other modifications:

In Fig. 2b, please show a matrix of % variable domain overlap, to complement the graphic provided.

Version 1:

Reviewer comments:

Reviewer #1

(Remarks to the Author)

The authors addressed or justified all my comments.

Reviewer #2

(Remarks to the Author)

Gieselmann and colleagues have responded robustly to my comments.

I like their updated description of the neutralization dependence of antibody 007 versus other V3 glycan antibodies shown in Fig. 4. One note that should be added to the analysis, however, is the fact that antibodies 10-1074 and PGT121 are from the same individual (and most likely the same antibody lineage), so this should be noted on the figure; in general, I agree with the authors' choices of comparator antibodies, as having both 10-1074 and PGT121 provides a nice control into the degree of neutralization divergence that can occur within a single lineage.

In terms of their response to my suggestion to compare neutralization between donor and 007, the correlation between donor neutralization and 007 neutralization is high. I commend the authors on the nice plot in Extended Data Fig. 1f. (As one additional control, it would be helpful – and quite easy to report - the same Extended Data Fig. 1g analysis not only for 007 but for other V3 glycan antibodies, to show that the 007/donor EN01 degree of correlations does not hold for other V3 glycan-broadly neutralizing antibodies; the authors should deposited their antibody neutralization data with CATNAP, so comparative analysis can be done.)

Other types of neutralization analysis that I suggested have been nicely done and integrated into the revised manuscript. I'm still puzzled by the Fab/IgG difference in neutralization, and feel the suggestion by the authors that intra-spike bi-valent binding occurs may have merit. In general the authors do a nice job trending the line between data and speculation.

Decision Letter:

Our ref: NI-A40406A

3rd Oct 2025

Dear Dr. Klein,

Thank you for submitting your revised manuscript "Identification of a broad and potent V3 glycan site bNAb targeting an N332 glycan-independent epitope" (NI-A40406A). It has now been seen by the original referees and their comments are below. The reviewers find that the paper has improved in revision, and therefore we'll be happy in principle to publish it in Nature Immunology, pending minor revisions to satisfy the referees' final requests and to comply with our editorial and formatting guidelines.

We will now perform detailed checks on your paper and will send you a checklist detailing our editorial and formatting requirements in about a week. Please do not upload the final materials and make any revisions until you receive this additional information from us.

If you had not uploaded a Word file for the current version of the manuscript, we will need one before beginning the editing process; please email that to immunology@us.nature.com at your earliest convenience.

Thank you again for your interest in Nature Immunology Please do not hesitate to contact me if you have any questions.

Sincerely,
Paula

Paula Jauregui, PhD
Senior Editor
Nature Immunology

Reviewer #1 (Remarks to the Author):

The authors addressed or justified all my comments.

Reviewer #2 (Remarks to the Author):

Gieselmann and colleagues have responded robustly to my comments.

I like their updated description of the neutralization dependence of antibody 007 versus other V3 glycan antibodies shown in Fig. 4. One note that should be added to the analysis, however, is the fact that antibodies 10-1074 and PGT121 are from the same individual (and most likely the same antibody lineage), so this should be noted on the figure; in general, I agree with the authors' choices of comparator antibodies, as having both 10-1074 and PGT121 provides a nice control into the degree of neutralization divergence that can occur within a single lineage.

In terms of their response to my suggestion to compare neutralization between donor and 007, the correlation between donor neutralization and 007 neutralization is high. I commend the authors on the nice plot in Extended Data Fig. 1f. (As one additional control, it would be helpful – and quite easy to report - the same Extended Data Fig. 1g analysis not only for 007 but for other V3 glycan antibodies, to show that the 007/donor EN01 degree of correlations does not hold for other V3 glycan-broadly neutralizing antibodies; the authors should deposited their antibody neutralization data with CATNAP, so comparative analysis can be done.)

Other types of neutralization analysis that I suggested have been nicely done and integrated into the revised manuscript. I'm still puzzled by the Fab/IgG difference in neutralization, and feel the suggestion by the authors that intra-spike bi-valent binding occurs may have merit. In general the authors do a nice job trending the line between data and speculation.
